# Mitochondrial Reactive Oxygen Species Dysregulation in Heart Failure with Preserved Ejection Fraction: A Fraction of the Whole

**DOI:** 10.3390/antiox13111330

**Published:** 2024-10-31

**Authors:** Caroline Silveira Martinez, Ancheng Zheng, Qingzhong Xiao

**Affiliations:** Centre for Clinical Pharmacology and Precision Medicine, William Harvey Research Institute, Faculty of Medicine and Dentistry, Queen Mary University of London, London EC1M 6BQ, UK; caroline.s.martinez@gmail.com (C.S.M.); a.zheng@qmul.ac.uk (A.Z.)

**Keywords:** heart failure, heart failure with preserved ejection fraction (HFpEF), cardiac diastolic dysfunction, cardiovascular disease, reactive oxygen species, mitochondrial, mitochondrial dysfunction, oxidative stress, redox signal

## Abstract

Heart failure with preserved ejection fraction (HFpEF) is a multifarious syndrome, accounting for over half of heart failure (HF) patients receiving clinical treatment. The prevalence of HFpEF is rapidly increasing in the coming decades as the global population ages. It is becoming clearer that HFpEF has a lot of different causes, which makes it challenging to find effective treatments. Currently, there are no proven treatments for people with deteriorating HF or HFpEF. Although the pathophysiologic foundations of HFpEF are complex, excessive reactive oxygen species (ROS) generation and increased oxidative stress caused by mitochondrial dysfunction seem to play a critical role in the pathogenesis of HFpEF. Emerging evidence from animal models and human myocardial tissues from failed hearts shows that mitochondrial aberrations cause a marked increase in mitochondrial ROS (mtROS) production and oxidative stress. Furthermore, studies have reported that common HF medications like beta blockers, angiotensin receptor blockers, angiotensin-converting enzyme inhibitors, and mineralocorticoid receptor antagonists indirectly reduce the production of mtROS. Despite the harmful effects of ROS on cardiac remodeling, maintaining mitochondrial homeostasis and cardiac functions requires small amounts of ROS. In this review, we will provide an overview and discussion of the recent findings on mtROS production, its threshold for imbalance, and the subsequent dysfunction that leads to related cardiac and systemic phenotypes in the context of HFpEF. We will also focus on newly discovered cellular and molecular mechanisms underlying ROS dysregulation, current therapeutic options, and future perspectives for treating HFpEF by targeting mtROS and the associated signal molecules.

## 1. Introduction

Heart failure (HF) is a potentially fatal medical condition characterized by a combination of cardiac and non-cardiac symptoms. Recent guidelines classify HF into distinct phenotypes based on left ventricular ejection fraction (LVEF): HF with reduced ejection fraction (HFrEF, LVEF as ≤40%), HF with mildly reduced ejection fraction (HFmrEF, LVEF between 41% and 49%), and HF with preserved ejection fraction (HFpEF, LVEF ≥ 50%) [1]. According to the latest ESC guideline, apart from having a LVEF ≥ 50%, the presence of symptoms and signs of HF (exertional intolerance, breathlessness, extravascular fluid accumulation in the lungs, subcutaneous tissues, and abdominal cavity), evidence of structural and/or functional cardiac abnormalities, and/or raised natriuretic peptides (NPs) are diagnostic indicators of HFpEF [1]. HFpEF is a significant issue in public health that imposes a substantial healthcare burden on both individuals and countries. The expenditure of HF-related costs in 2012 is estimated to be USD 30.7 billion, and the projection is an increase of 127% by 2030 [1,2]. HFpEF is not an isolated cardiac disease but a metabolic syndrome with a high comorbidity burden, with its prevalence increases with age. Common comorbidities include hypertension, diabetes, obesity, atrial fibrillation (AF), anemia, chronic obstructive pulmonary disease (COPD), and chronic kidney disease (CKD). The combination of these comorbidities seems to play a key pathophysiological role in the development of the myocardial stiffening, fibrosis, and diastolic dysfunction via systemic microvascular inflammation and coronary microvascular dysfunction in patients with HFpEF [3].

Increasing evidence has suggested that excessive reactive oxygen species (ROS) generation and oxidative stress seem to play a critical role in the pathogenesis of HFpEF [4,5]. Although ROS are normally linked to harmful effects on cardiac remodeling, small amounts of ROS is required for maintaining mitochondrial homeostasis and cardiac functions. In this review, we will provide an overview and discuss recent findings on mitochondrial ROS (mtROS) production, its threshold for imbalance, as well as consequent dysfunction causing related cardiac and systemic phenotypes in the context of HFpEF, focusing on newly discovered cellular and molecular mechanisms underlying ROS dysregulation, the current therapeutic options, and future perspectives to treat HFpEF by targeting mtROS and associated signal molecules.

## 2. HFpEF

### 2.1. Worldwide Prevalence, Morbidity, and Mortality

HF affects more than 64 million people worldwide [6]. In Europe, the prevalence of HF is estimated to be 1% for those aged <55 and >10% for those aged 70 years or over [1]. Epidemiological research with LVEF registries (2000–2016) indicates that HFpEF accounts for as much as 55% of all HF incidence [7]. The rising occurrence of HFpEF may be attributed in part to the aging population, heightened risk factors, and related comorbidities, as well as advancements in diagnostic accuracy [7]. Unfortunately, no drug has been identified to reduce mortality and morbidity in HFpEF so far. The available treatments seem to be more responsive in patients with HFmrEF and HFrEF [1]. Additionally, comorbidities including cardiovascular (CV) and non-CV diseases also significantly influence the treatment response and outcomes of HFpEF. Among patients with HF, there is a rise in non-CV comorbidities as the population ages. In HFpEF, the co-presence of CKD, COPD, depression, sleep disorders, obesity, diabetes mellitus, cancer, and iron deficiency all can hinder the diagnosis and have a great influence on clinical phenotype, response to drugs, hospitalization, prognosis, and health-related quality of life [8]. Alarmingly, the prevalence of HFpEF has risen from 38% to 55% among individuals hospitalized for HF during the past 35 years [9], and hospital admissions for HF are predicted to increase up to 50% in the next 25 years as a result of the aged population and the increased incidence of comorbidities [1,10]. Prognostic studies for individuals with HFpEF can differ based on factors such as the design of the study, the presence of other medical conditions, and the stage of the disease. Observational studies indicate that patients with decompensated HFpEF had mortality rates of approximately 20–30% within one year [9,11,12,13,14] and, after first hospitalization, the short-term mortality of HFpEF is comparable with HFrEF [9].

The primary cause of death for HFpEF patients is variable according to the nature of the study and the characteristics of the population. For instance, it has been reported that CV diseases (CVDs) are the main cause of death varying from 60% to 70% in clinical trials [15], and patients with older age groups show lower rates of CVDs as the cause of death, varying from 50 to 60% [15]. Interestingly, the incidence of non-CV fatalities is typically greater in patients with HFpEF compared to those with HFrEF [16]. Corroborating with this finding, a report from Olmsted County showed a rate of 49% of non-CV deaths in HFpEF patients [12]. Moreover, according to the Framingham Heart Study, the non-CV mortality rates may vary depending on sex, with 61% in men and 51% in women [17]. Sudden death accounts for approximately 25% of all-cause mortality in HFpEF [15]. A recent long-term follow-up study shows that after 2.1–7.9 years of an acute decompensated HFpEF, 68% of patients died, with 47% from CV and 45% from non-CV causes. Specifically, data showed that coronary artery disease (CAD) and tricuspid regurgitation were predictors of CV death, and stroke, kidney disease, lower BMI, and lower sodium were predictors of non-CV death, respectively. Anemia and higher age were related to both events [18]. Similarly, a recent observational and retrospective analysis showed that anemia and iron deficiency significantly increase mortality risk among HFpEF [19]. In this study, anemia, advanced age, iron deficiency, decreased LVEF, CKD, and paroxysmal nocturnal dyspnea were related to all-cause mortality in HFpEF [19].

### 2.2. The Pathophysiology of HFpEF

HFpEF does not have a cardiocentric physiopathology, with heterogeneous clinical presentation attributing to different comorbidities from several organ/tissue systems [20] (Figure 1). There has been recent recognition that comorbidities such as diabetes, obesity, and renal, pulmonary, or skeletal muscle diseases, often associated with inflammation and visceral fat, contribute to the development and outcomes of HFpEF and stimulate the creation of the term “inflammatory-metabolic HFpEF”, as recently reviewed [21]. Macrophage infiltration induces adipose tissue inflammation, upregulating proinflammatory adipokines (e.g., leptin, TNF-α, IL-6, and resistin) and downregulating anti-inflammatory adipokines (e.g., adiponectin, omentin-1), resulting in chronic and low-grade systemic inflammation [22,23,24]. The activation of these proinflammatory pathways seems to raise systemic inflammation and oxidative and nitrosative stress, which cause mitochondrial and endothelium dysfunctions, contributing to multisystemic abnormalities in HFpEF [21].

Diastolic dysfunction plays a key role in the pathophysiology of HFpEF, but there are also multiple cardiac, vascular, and extracardiac problems that contribute to this condition. The pathophysiology of HFpEF encompasses deficiencies in the relaxation and contraction abilities of the left ventricle (LV), abnormalities in the functioning of the left atrium (LA), high pressure in the lungs (pulmonary hypertension or PH), abnormalities in the exchange of gases, dysfunction in the right side of the heart, irregularities in the autonomic nervous system, stiffening of blood vessels, inadequate blood supply to the heart muscle (myocardial ischemia), impaired functioning of the endothelium, kidney disease, and issues in the peripheral tissues such as skeletal muscle and fat [4]. In the following sections, we will discuss the key pathophysiologic mechanisms of HFpEF, with a primary emphasis on the heart, as most of the available material pertains to this organ.

Diastolic dysfunction is produced by an elevation in passive myocardial stiffness due to changes in the viscoelastic properties of the heart muscle [25]. The stiffness of the LV alters the relationship between end-diastolic pressure and volume, which will be more prominent during physical activity [26,27]. The primary molecular cause of LV stiffness is the protein titin, which acts as a two-way spring in cardiomyocytes (CMs) and is regulated by phosphorylation through cyclic guanosine monophosphate (cGMP) kinases [28]. The phosphorylation process is affected by the presence of nitric oxide (NO), linking the health of blood vessels to the functioning of the heart [29]. Moreover, alterations in the extracellular matrix, specifically the augmentation of fibrillar collagen, play a role in the rigidity of the heart muscle and are linked to elevated myocardial fibrosis [30].

HFpEF patients may present systolic dysfunction, which commonly occurs due to issues associated with calcium control, beta-adrenergic signaling, myocardial energy utilization, or tissue blood flow [25]. Although these systolic dysfunctions may be moderate when the body is at rest, they can worsen when subjected to physical stress, resulting in a decrease in the amount of blood pumped by the heart and an increase in the inability to tolerate exercise [31,32]. Systolic dysfunction not only impairs cardiac efficiency but also worsens diastolic dysfunction by limiting the heart’s capacity to contract effectively and decreasing the ventricular elastic recoil required for adequate diastolic filling. The abovementioned sequence of cardiac malfunctions can result in higher death rates among patients with HFpEF and also contribute to other comorbidities such as PH and LA dysfunction due to decreased atrioventricular coupling, as nicely documented in the literature [33,34,35].

The LA initially counterbalances LV diastolic failure by functioning as a blood storage area, facilitating LV filling without raising LA pressures [20]. Over time, chronic or severe LV dysfunction results in the enlargement and impaired function of the LA, which is strongly associated with the progression of PH, right ventricular dysfunction (RVD), and AF [36,37]. AF is identified as a notable indicator of LA myopathy [37]. It worsens diastolic dysfunction by elevating the burden on the atria and interrupting the synchronization between the atria and ventricles [37]. As a result, AF contributes to mitral regurgitation and overall inefficiency of the heart. This series of events leads to an increase in diastolic ventricular interaction (DVI), where the enlargement of the LA and right side of the heart increases the overall volume of the heart [37,38]. This puts pressure on the pericardium, which disrupts the Frank–Starling mechanism that is crucial for maintaining the output of blood from the heart [38]. Echocardiographic techniques, such as speckle tracking, reveal that LA strain and compliance gradually decline due to persistent LV diastolic dysfunction [36,39,40]. This loss in cardiac function not only distinguishes HFpEF from other factors causing difficulty in breathing but also increases the likelihood of developing AF, resulting in worse clinical results such as higher mortality rates, reduced ability to engage in physical activity, and worsened RVD [36,37,41,42].

In addition, a significant percentage (between 50 and 80%) of patients with HFpEF develop PH [43]. This is mainly caused by increased pressure in the LA, leading to isolated postcapillary PH. Prolonged exposure to these increased pressures exacerbates the resistance and consequent remodeling of the blood vessels in the lungs, resulting in a combination of pre- and postcapillary PH [43]. This condition not only reduces the ability to exercise but also raises the likelihood of right HF, hospitalization, and death [44].

The right ventricle (RV), which is not well adapted to handle high afterload, experiences severe negative effects under these circumstances. This leads to systemic venous congestion and a variety of problems including edema, organ failure, AF, and cardiac cachexia [45,46,47]. Studies emphasize that RVD in HFpEF is substantially associated with poorer outcomes [41,47]. In addition, prolonged elevated LA pressure harms pulmonary capillaries, impairing lung diffusion and the exchange of carbon monoxide, which in turn increases mortality risk in HFpEF patients [48,49]. Moreover, HFpEF is characterized by hypertension in about 90% of patients, with a high prevalence of increased stiffness in the aorta and conduit vessels, especially in individuals with diabetes [50,51]. Blood pressure (BP) instability occurs as a result of the combined effects of ventricular and arterial stiffness [52]. Patients diagnosed with HFpEF also demonstrate impaired vasodilation that depends on the endothelium and is related to systemic inflammation and a decrease in the availability of NO [3,53,54]. Moreover, there is a high incidence of coronary microvascular dysfunction, which impacts both endothelium-dependent and independent mechanisms [55]. This dysfunction is linked to severe disease indicators such as AF, RVD, and limited exercise capacity [56,57,58].

While the cardiac output stays within a normal range during periods of rest, the potential to enhance it during physical effort is compromised, leading to a decrease in aerobic capacity [59]. This can be attributed in part to a reduction in stroke volume (SV) reserve caused by cardiac dysfunction and chronotropic incompetence [60]. The restriction in heart rate reserve appears to be associated with a decrease in adrenergic sensitivity rather than problems with central outflow [60,61].

Research on autonomic function in HFpEF yields conflicting findings; certain studies suggest impaired arterial baroreflex sensitivity [60], while others observe intact parasympathetic and sympathetic responses to stress [62]. In rodent models, a reduction in baroreflex sensitivity results in challenges in regulating variations in volume, which in turn leads to an elevation in left atrial pressure [63]. Sympathetic outflow aids in narrowing large-capacity veins, increasing the flow of blood back to the heart by moving it from less strained to more strained areas. This process, which is especially evident in HFpEF associated with obesity, plays a key role in the elevation of ventricular filling pressures during physical activity [32].

Individuals with HFpEF frequently demonstrate reduced oxygen consumption (VO_2_) as a result of irregularities in the differential in oxygen content between arteries and veins (AVO_2diff_). This indicates problems in the delivery and utilization of oxygen in the skeletal muscles [64,65]. Histological examinations reveal a decrease in the number of blood vessels and a change towards a higher proportion of Type II (fast-twitch) muscle fibers, with a lower number of Type I (aerobic) fibers [66]. Additionally, there is evidence of impaired mitochondrial function, which further hampers the utilization of oxygen in the peripheral tissues [67].

## 3. Animal Models of HFpEF

Multiple animal models have been reported to simulate human HFpEF in the past decades, and each comes with unique advantages and disadvantages (Table 1). The first animal models used to investigate HFpEF focused on hypertension, diastolic dysfunction, or left ventricular hypertrophy (LVH), such as pressure overload in Dahl salt-sensitive rats (DSSRs) [68,69] and spontaneously hypertensive rats (SHRs) [70], as well as models that involve renovascular hypertension, aortic constriction, or hormone-induced hypertension [71]. These models have been used in the investigation of the fundamental mechanisms of HFpEF, with a particular emphasis on hypertension and CV stress. However, there are two major limitations: First is that these animals respond to treatment with angiotensin II or another hormone blockade [72,73], which does not happen in the human syndrome. Second, these models show two different phases that normally do not occur in human disease [74]. Initially, there is a hypercompensated phase characterized by preserved LVEF, delayed diastolic relaxation, and impaired ventricular compliance [75]; after this initial phase, there is a more “dilated period” where both systolic and diastolic BP decrease [69,76]. Despite these limitations, models of pressure–volume overload are still good strategies for testing new treatments for HFpEF [77,78].

With the global spread of cardiometabolic HFpEF, animal models combining hemodynamic and metabolic stress have been developed [21,79]. One good example is the crossing of the spontaneously hypertensive HF rat with the Zucker diabetic rat, carrying a leptin receptor mutation (Lepr), generating the ZSF1 rat [79]. The generated ZSF1 animal model exhibits a combination of diastolic dysfunction, hypertension, LVH, cardiac fibrosis, obesity, hyperlipidemia, renal dysfunction, reduced NO signaling, and aortic stiffening—key characteristics of HFpEF [80]. Despite not developing fluid imbalances, the ZSF1 rat has become widely and successfully adopted in both academic and industry research as a model for studying HFpEF [81,82]. However, there are some limitations to using the ZSF1 rat model for HFpEF. The ZSF1 rat does not present increased LVDP or raised natriuretic peptide until a very later stage and responds to the inhibition of the angiotensin-converting enzyme [83]. Moreover, due to the mixed genetic background, there is a dilution of each phenotype behind it.

Another often-used animal model is the two-hit model, which combines the inhibition of nitric oxide synthase 1 and 3 (NOS1 and NOS3) by N-nitro-l-arginine methyl ester (l-NAME) with a high-fat diet in C57BL/6 mice [5]. The two-hit model is well defined, yielding robust and dependable findings from assessments of both inactivity and physical exertion. This model gives us important information regarding cardiac histology, exercise intolerance, and physiopathology of HFpEF, crucial for studying the human syndrome. However, it has some important limitations. First, female mice are less susceptible to developing the HFpEF phenotype in this model [84], which contrasts with the human situation where postmenopausal women are strongly impacted by HFpEF. Second, the blockage of NOS by l-NAME to induce hypertension does not represent the underlying mechanisms of hypertension in HFpEF patients. Lastly, the two-hit model does not show skeletal muscle alterations [5], therefore limiting its ability to fully represent human disease.

The use of larger mammals such as dogs, pigs, and non-human primates can increase therapeutic applicability and translation in the study of HFpEF. A dog model of secondary hypertension because of kidney disease leads to cardiac responses similar to HFpEF syndrome [85,86]. Similarly, pig models receiving a combination of deoxycorticosterone acetate (DOCA), a Western diet, and salt shows important features of HFpEF [87,88,89]. The pig model of HFpEF shows an enlargement of the LA with decreased contraction capacity, scarring of the heart muscle, increased oxidative and nitrosative stress, and reduced titin phosphorylation [87,88,89]. Nevertheless, these models still fail to completely replicate the intricacies of the human situation. In comparison to HFpEF patients, the pig models have lower diastolic pressure, increased relaxation time, and a smaller LV chamber with compensatory hypertrophy [87,88].

Indeed, the creation of a HFpEF animal model that effectively replicates the entire range of severity and comorbidities of HFpEF as seen in humans is a big challenge. The models that attempt to simulate the cardiometabolic phenotype (e.g., obesity + hemodynamic alterations) have the biggest potential. However, neither of them can properly simulate obesity or hypertension as observed in HFpEF patients. These patients usually show obesity grade 2 or higher (BMI ≥ 35 kg/m^2^, WHO classification), which is difficult to achieve with animal models. Regarding BP, patients often have treated BP (systolic BP < 130 mmHg) while, in the animal models, the hypertension normally remains untreated. In addition, fluid retention, skeletal muscle alternations, and metabolic dysfunction should be encouraged in animal models of HFpEF. Nonetheless, to evaluate prospective medicines and achieve translational relevance, easily implemented animal models should be used, including large mammals, even with the high cost and absence of the full reproducibility of the syndrome.

## 4. Mitochondrial Metabolism in Cardiac Hemostasis and Injury-Induced Remodeling

### 4.1. Mitochondria—The Powerhouse of the Heart

Human individuals produce and consume an amount of adenosine triphosphate (ATP) nearly equivalent to their body weight every single day [90]. The heart, in particular, has high energy demands, accounting for only 0.5% of body weight but consuming roughly 8% of ATP produced. Moreover, mitochondria occupy approximately one-third of the volume of adult CMs and produce the majority of the ATP consumed by the heart (~95%) [91,92,93]. Mitochondria produce abundant ATP by metabolizing a variety of fuels including fatty acids (FAs), glucose, lactate, ketones, pyruvate, and amino acids, primarily by mitochondrial oxidative phosphorylation (OXPHOS) (Figure 2). Any disruptions in the metabolic energy pathways that produce ATP can have catastrophic consequences on cardiac functions, resulting in a variety of CVDs including HFpEF [94]. Mitochondria are organelles with a double membrane structure, containing an inner and an outer membrane (Figure 2). The inner membrane of mitochondria is convoluted into many infoldings called cristae. Mitochondrial fragmentation and cristae destruction were clearly evident, with decreased mitochondrial area in HFpEF [95]. OXPHOS primarily consists of five enzymatic complexes, namely the electron transport chain (ETC), carried out by large multi-subunit protein complexes in the cristae membranes [96], which produce ATP by coupling the oxidation of reducing equivalents in mitochondria to the generation and subsequent dissipation of a proton gradient across the inner mitochondrial membrane [97].

The tricarboxylic acid (TCA) cycle, also known as the citric acid cycle or the Krebs cycle, is localized in the inner mitochondria matrix and is a closed loop that forms a metabolic engine within cells [98]. The TCA cycle itself does not consume molecular oxygen or produce meaningful amounts of ATP (Figure 2). However, it mainly transfers electrons from inputs (e.g., acetyl-CoA) to electron carriers (e.g., NAD^+^, FAD^+^), which then go into ETC and produce abundant ATP [99]. From an energy-generating perspective, the primary function of the TCA cycle is to oxidize acetyl-CoA to generate two molecules of CO_2_, two ATP, and some byproducts including three NADH and one FADH2. These byproducts further feed into the ETC at complex I (NADH dehydrogenase) and complex II (succinate dehydrogenase), respectively, to ultimately produce 30–32 ATP molecules through OXPHOS. Importantly, TCA cycles also produce and release abundant metabolites to control the biosynthesis of macromolecules such as nucleotides, lipids, and proteins, playing a critical role in cellular homeostasis, stress response, and disease pathogenesis [100,101].

### 4.2. Mitochondrial Metabolism and Metabolites in Cardiac Hemostasis

In the adult heart, ATP produced through OXPHOS in mitochondria accounts for 95% of myocardium ATP requirements, while 5% of ATP comes from anaerobic glycolysis [102] (Figure 2). In oxidative metabolism, the majority of ATP production (around 60%) is from the oxidation of mitochondrial FAs, followed by lesser contributions from glucose, lactate, and ketone bodies [103]. Changes in substrate availability and uptake can happen in the healthy heart to maintain ATP production [104]. For instance, under well-fed conditions, glucose can become a significant fuel for the heart via glycolysis and glucose-derived pyruvate oxidation. However, in fasting conditions, FAs can serve as the primary cardiac fuel.

FAs can enter CMs to produce fatty acyl-CoA, which is then transported into the mitochondria. Fatty acyl-CoA undergoes β-oxidation to produce acetyl-CoA, serving as an important substrate for the TCA cycle [105]. Glucose is also an important fuel for the heart because it can generate ATP both from anaerobic glycolysis and providing pyruvate to the TCA cycle. The majority of pyruvate derived from glycolysis is converted to acetyl-CoA by PDH (pyruvate dehydrogenase), and acetyl-CoA is then further metabolized in the TCA cycle. Additionally, lactate is increasingly being recognized as an important energy substrate of the heart. Lactate can be produced and released to the blood by skeletal muscle [106], then taken up by the heart where it converts into pyruvate via LDH (lactate dehydrogenase). The resultant pyruvate is then converted to acetyl-CoA, which enters into the TCA cycle in a similar way to the pyruvate generated from glycolysis. Recent studies also found that under some circumstances, lactate can provide carbons to the TCA cycles in a pyruvate-independent manner [107]. Moreover, it has been proved that ketone bodies are also important fuel in the heart. The ketone bodies, namely acetoacetate, acetone, and β-hydroxybutyrate (βOHB), are produced in the liver, with βOHB being the predominant ketone body taken up and oxidized in the heart [108,109]. After uptake, βOHB is transported to mitochondria and is oxidized to acetoacetate by BDH1 (β-hydroxybutyrate dehydrogenase 1), which is then converted to acetoacetyl-CoA (Acetoacetate-CoA) and undergoes a thiolysis reaction to produce acetyl-CoA. If circulating levels of ketone bodies are elevated, they can become a major fuel in the heart [110]. Finally, recent evidence shows that amino acids are also a potential source in ATP production in the heart. Branched chain amino acid (BCAA) oxidation, which occurs in the mitochondria, is the best characterized source of amino acid oxidation in the heart. BCAAs will be converted to branched chain α-keto acids (BCKAs) via reversible transamination, followed by irreversible decarboxylation by the branched chain α-keto acid dehydrogenase (BCKDH), eventually generating acetyl-CoA for the TCA cycle and succinyl-CoA for anaplerosis, respectively [111].

Importantly, TCA cycle metabolites appear to be an important mitochondrial messenger, capable of eliciting both genetic and epigenetic reprogramming, as exemplified by acetyl-CoA produced via pyruvate conversion, FA oxidation (FAO), and ketone body oxidation [112] (Figure 2). Acetyl-CoA is a building block for the biosynthesis of lipids, ketone bodies, amino acids, and cholesterol. It also provides acetyl groups for histone acetylation and modulates the activity of metabolic enzymes [113]. Another example is succinate. Under aerobic conditions, succinate is mainly produced from succinyl coenzyme A and converted to fumarate by the TCA cycle enzyme succinate dehydrogenase (SDH). Succinate can act as a messenger between intracellular metabolic status and cellular functions to regulate DNA methylation and lysine succinylation. Under low oxygen conditions, SDH activity is reduced, causing succinate accumulation and decreased fumarate [114,115]. Fumarate converted from succinate also has an important role in chromatin modifications and protein succination [116]. Therefore, TCA cycle metabolites not only function as a part of the TCA cycle and contribute to the production of ATP but also play important roles in biosynthesis of macromolecules, as well as genetic and epigenetic regulation.

### 4.3. Mitochondrial Metabolism and Metabolites in Injury-Induced Cardiac Remodeling, with a Focus on HFpEF

Once ATP synthesis is stopped, the stored ATP can only sustain the heartbeat for a few seconds [117]. Therefore, the rate of ATP synthesis must match the rate of ATP consumption in the heart, and undoubtedly the most crucial change happening in mitochondrial energy metabolism observed in the failing heart is energy or ATP deficiency [118] (Figure 3). The metabolic changes begin during compensated cardiac hypertrophy and are accentuated in overt HF [119]. As mentioned before, a healthy heart uses FAs as its primary energy fuel to produce ATP, while glucose becomes the main energy fuel in HF, as evidenced by increased glucose uptake and glycolysis but decreased FAO and OXPHOS in HFrEF [120,121]. Similar to HFrEF, HFpEF myocardium has reduced expression levels of proteins involved in FA uptake and oxidation, indicating impaired utilization of FA in the heart [122,123,124]. Unlike in HFrEF, glucose uptake and metabolism are inhibited in HFpEF myocardium [122,125]. Instead, HFpEF patients show increased amino acids in the heart and circulating blood, and the main upregulated amino acids in the HFpEF heart are BCAAs. However, their catabolites are lower compared to healthy myocardium, indicating impaired BCAA oxidation in HFpEF myocardium [122,126]. Ketone body oxidation is considered protective for HFrEF since it provides an important alternative fuel for ATP production in the heart. Compared to the control, no apparent changes in ketone body amount are found in the HFpEF heart [122]. However, it has been shown that the serum concentrations of the ketone bodies acetoacetate, 2-hydroxybutyrate, and 3 hydroxybutyrate are lower in HFrEF patients than both HFpEF patients and non-HF controls [126], implying a greater reliance on ketone bodies as an energy source in HFrEF patients. Interestingly, new evidence from recent clinical trials with sodium-glucose co-transporter-2 (SGLT2) inhibitors, which can increase circulating ketones, proves the beneficial effects of SGLT2 inhibitors on human HFpEF [127,128]; however, the underlying mechanisms remain unknown.

The cardiac expression levels of pyruvate and acetyl-CoA in HF are increased, indicating inadequate TCA cycle intermediate replacement. Lower levels of TCA cycle intermediates such as malate, fumarate, and succinate in HFpEF myocardium also suggest inadequate anaplerosis [122]. PDH converts pyruvate to acetyl-CoA in mitochondria, and its activity varies inversely with protein phosphorylation by PDK4. Both phospho-PDH/total PDH ratio and expression of PDK4 were significantly lower in HFpEF, which would favor enhanced pyruvate conversion to acetyl-CoA [129]. Except for contributing to producing ATP, acetyl-CoA and TCA cycle metabolites also have important roles in post-translational modifications, especially in acetylation. Protein acetylation is increased in failing human and mouse hearts, which is more exacerbated in HFpEF compared with HFrEF [123,130,131]. Hyperacetylation has been linked with decreased enzymatic activities of multiple mitochondrial proteins and impaired FAO [123,132]. Sirtuin enzyme Sirt3 knockout mice even developed more pronounced diastolic dysfunction, suggesting a protective effect of CM Sirt3 in HFpEF pathogenesis [123]. NAD^+^ repletion via oral supplementation of an NAD^+^ precursor nicotinamide riboside in mice reversed the hyperacetylation of FAO key enzymes and thus restored mitochondrial function and improved pathological cardiac remodeling in the HFpEF model [123]. This increased protein acetylation could be attributed to the enlargement of the acetyl-CoA pool in the mitochondrial matrix [133]. Interestingly, the administration of ketone body β-hydroxybutyrate can downregulate the acetyl-CoA pool and protein acetylation, partially via activation of citrate synthase and inhibition of FA uptake in HFpEF mice [133].

Genetic studies unravel the genetic links between mitochondrial metabolism and cardiac pathologies. SDH (succinate dehydrogenase) can link the TCA cycle with the ETC, therefore playing a unique role in mitochondrial metabolism. Dilated cardiomyopathy (DCM) can be caused by a G555E mutation in the SDHA gene, which significantly reduces tissue-specific SDH enzymatic activity in cardiac muscle [134]. Mice with cardiac deletion of SDHAF4 (succinate dehydrogenase assembly factor 4), an assembly factor that is responsible for proper assembly of the SDH subunits [135], display SDH activity suppression, metabolic deficiency, and DCM phenotypes. Importantly, supplementation with fumarate, which is depleted due to insufficient SDH activity, in SDHAF4 knockout mice can substantially improve mitochondrial dynamics and cardiac functions, as well as prolong the lifespan of DCM mice [136]. These findings show that the normal function of SDH is important for producing enough fumarate and maintaining normal cardiac physiological functions. However, the potential functions of SDH and fumarate need to be further studied in HFpEF.

A recent study showed that mitochondrial complex disorder may occur from the early stages of HFpEF, evidenced by reduced expression of key proteins required for complex I, ATP synthase, efficient electron transfer, and supercomplex formation [137]. Complex I-mediated mitochondrial respiration is impaired in permeabilized cardiac fibers of Zucker/spontaneously hypertensive F1 hybrid obese (ZSF1) rats that develop an HFpEF-like phenotype. The suppression of mitochondrial respiration could, in part, be caused by increased mitochondrial matrix Ca^2+^ levels, which would contribute to a lower proton motive force for ATP generation [138,139]. Mitochondrial complex I deficiency by Ndufs4 gene inactivation increases mitochondrial protein acetylation and accelerates HFrEF [140]. But the function of complex I in HFpEF has not been well studied. Limited studies using patients’ cardiac tissues showed that genes involved in OXPHOS are elevated in HFpEF but decreased in HFrEF. However, the higher level of OXPHOS expression in HFpEF was attenuated after adjustment for body mass index, suggesting that obesity may contribute to the elevated OXPHOS gene expression in HFpEF [141]. It is worth mentioning that studying cardiac tissues from patients with HFpEF is extremely limited, likely due to challenges in obtaining cardiac tissue from HFpEF patients, as opposed to patients with HFrEF.

PCSK9 (proprotein convertase subtilisin/kexin type 9) is known to be secreted into the circulation, mainly by the liver. It interacts with low-density lipoprotein receptor (LDLR) homologous and non-homologous receptors, thus favoring their intracellular degradation [142]. Therefore, PCSK9 inhibitors have been widely used to treat hypercholesterolemia and to reduce cardiovascular risk in patients with ischemic heart disease. Interestingly, it has been reported that PCSK9 genetic deficiency promotes HFpEF progression, independent of the modulation of LDLR and of circulating PCSK9 [143]. PCSK9-deficient mice display impaired OXPHOS and mitochondrial metabolism, evidenced by impaired FAO and decreased TCA cycle flux. Such impairments may be attributed to increased cholesterol accumulation in the heart [143]. This new evidence raises caution in terms of the safety issue with the increasing use of PCSK9 inhibitors in clinics.

## 5. Mitochondrial Quality Control (MQC): The Protective “Mito Orchestra” in the Context of HFpEF

As mentioned in the above section, mitochondria are vital organelles to maintain cardiac homeostasis. CMs are dependent on ATP production via OXPHOS to fulfill the energy requirements of the organ. Therefore, the integrity of mitochondrial function is achieved through the implementation of an orchestrated MQC mechanism [144] (Figure 4). Disruptions in the MQC mechanisms have detrimental effects on the function and contractility of CMs, potentially leading to cellular apoptosis and tissue injury [145,146]. Indeed, mitochondrial dysfunction has a significant role in the pathogenesis of several CVs, such as hereditary cardiomyopathies, ischemia/reperfusion injury, hypertension, atherosclerosis, diabetes, and HF [147,148].

There are two levels of MQC mechanisms: molecular and organelle-level control. These orchestrated mechanisms are essential for the maintenance of mitochondrial function [149]. Mitochondrial chaperones, including heat shock proteins (HSP60, HSP70, and HSP90), are essential for the appropriate folding of proteins at the molecular level. These chaperones aid in the proper folding of newly synthesized proteins and the refolding of damaged ones before they are imported into the mitochondria. The ubiquitin–proteasome system (UPS) will then target proteins that are misfolded or damaged for degradation, trying to prevent the accumulation of dysfunctional proteins [150,151]. Organelle-level MQC mechanisms are activated when molecular-level mechanisms are insufficient to repair mitochondrial damage. These mechanisms address more severe mitochondrial dysfunction, assuring the maintenance of cellular health by addressing damaged mitochondria that cannot be repaired at the molecular level [152].

### 5.1. Mitochondrial Biogenesis

Mitochondrial biogenesis, the process by which cells increase mitochondrial mass, is essential for the preservation of cardiac homeostasis, particularly in the face of cellular injury [153,154]. By increasing mtDNA copy number and metabolic enzyme expression, mitochondria boost their metabolic capability. Mitobiogenesis involves a coordinated interaction between mitochondrial and nuclear genomes and is affected by several endogenous or exogenous stimuli such as cold, exercise, caloric restriction, cell division, and oxidative stress [155].

The transcriptional coregulator PPARγ coactivator 1α (PGC1α) is the main regulator of the mitobiogenesis process [156]. In response to external stimuli such as cold exposure, exercise, and fasting, PGC1α activates a variety of transcription factors such as Nuclear Respiratory Factors 1 and 2 (NRF1, 2) [157,158]. NRF1 or NRF2 induce mitochondrial transcription factor A (TFAM), which translocates to mitochondria where it transcribes mtDNA [159]. Moreover, mitochondria-resident microRNAs (miRs) assist in the regulation of the mitochondrial pool by targeting TFAM and other biogenesis factors, thereby increasing the mtDNA copy number and inducing the production of proteins that are essential for mitochondrial function [160,161,162].

The therapeutic potential of regulating the PGC1α signaling pathway is applicable to a variety of diseases, such as diabetic cardiomyopathy, obesity, and HF [163,164]. Indeed, PGC1α/β-deficient mice exhibit important mitochondrial structural dysregulation, including elongation and breakage, which results in fatal cardiomyopathy, small hearts, and reduced cardiac output [165,166]. In the context of HF, microarray analysis showed that cardiomyopathy-derived HF patients have reduced cardiac PGC1α expression compared to healthy individuals [167]. Similarly, in ischemic and non-ischemic animal models of cardiomyopathy (ICM and non-ICM), the PGC1α expression seems to be downregulated. The reduced PGC1α expression impacted mitochondrial content and mtDNA replication, particularly in the late phases of HF [168,169].

Nevertheless, the role of PGC-1α in HF needs to be fully addressed. Pereira et al. [170] showed in HF models that PGC1α overexpression did not improve myocardial contractility or mitochondrial function. Additionally, in transgenic mice with cardiac-specific overexpression of PGC-1α, excessive mitochondrial proliferation was observed, which was accompanied by structural disarray in the sarcoplasmic reticulum; this resulted in diminished myocardial contractility, cardiac enlargement, and non-ICM HF [171]. These studies agree with human studies that did not find changes in cardiac PGC1α expression in HF patients [172]. Specifically, the reduced PGC1α levels and complex IV activity were only observed in HFrEF [96]. This implies that the impact of PGC1α on HF may be contingent upon its level of expression, its interactions with other mitochondrial biogenesis effectors, and the broader intracellular environment.

### 5.2. Mitochondrial Dynamics

Mitochondria are very active organelles; through mitochondrial dynamics, they form a dynamic network driven by mitochondrial cytoskeleton movements, mitochondrial fusion and fission, and mitophagy [173,174,175]. Proteins located in the outer mitochondrial membrane (OMM), such as dynamin-related GTPases, also known as mitofusins (MFN1 and MFN2), and proteins located in the interior mitochondrial membrane (IMM), such as optic atrophy protein 1 (OPA1) and cardiolipin, are involved in mitochondrial fusion by which two mitochondria are fused sharing mtDNA, metabolites, and enzymes [176,177,178]. The fusion of the OMM initiates by homo- and hetero-dimerization of MFN1 and MFN2 while OPA1 drives the fusion of the IMM maintaining the cristae structure and normal membrane permeability, essential protection against oxidative damage [179].

The process of fission involves the separation of healthy mitochondria from a damaged mitochondrion [180,181]. The most important classes of proteins involved in the fission processes are dynamin-related protein 1 (DRP1), located in mitochondrial-associated membranes (MAMs), where mitochondrial fission primarily occurs. During the process, this GTPase is recruited to the OMM, where it interacts with receptors such as mitochondrial fission 1 protein (FIS1) and mitochondrial fission factor (MFF) to induce mitochondrial cleavage. Apart from acting as a receptor for DRP1, FIS1 can also inhibit fusion, then promote fission [182,183]. Furthermore, the endoplasmic reticulum (ER) can encircle mitochondria, which aids in the constriction of the membrane during fission [184]. Importantly, mitophagy is linked to fission, as it guarantees the elimination of dysfunctional mitochondria, thereby preserving the health of the mitochondrial population and cellular homeostasis [185,186].

Imbalances in mitochondrial fission and fusion seem to be related to cardiac hypertrophy and the development of HF. It has been shown that cardiac deletion of MFN1/MFN2 leads to an eccentric ventricular remodeling while DRP1 absence causes dilated cardiomyopathy in cardiac genetic-modified mice [187]. Moreover, MFN2 downregulation was observed in a rodent model of cardiac hypertrophy [188], and MFN2 upregulation could constrain myocardial hypertrophy [189,190]. A study also suggests a role for OPA1 in HF progression. In a HF mouse model of pressure overload with HFD feeding, OPA1 was found to facilitate energy metabolism and increase cardiac FA utilization, therefore reducing ROS production, mitochondrial fragmentation and cardiac dysfunction [191]. Interestingly, it has been recently proposed that the potential consequences of an imbalance between mitochondria fusion and fission may be more severe than the simultaneous cessation of both processes. This idea is supported by Song et al. (2017), which has demonstrated that MFN1/MFN2/DRP1 triple knockout mice exhibit a unique form of cardiac hypertrophy and longer survival time when compared to MFN1/MFN2 or DRP1 cardiac knockout mice [192].

Indeed, the dynamics of mitochondria, particularly the equilibrium between fission and fusion, seem to be dysregulated in HFpEF. While the number of studies on this subject is low, new evidence indicates a transition towards higher levels of mitochondrial fission in HFpEF. Specifically, increased DRP1 levels were found in LV tissues of HFpEF patients compared to HFrEF or healthy individuals, with limited changes in the fusion proteins OPA1 and MFN2 [96]. Moreover, abnormal mitochondria and reduced MFN2 levels were observed in the skeletal muscle of elderly HFpEF patients, which was related to a decreased peak VO2 and 6 min walk distance when compared to HFrEF or control individuals [193]. In line with this finding, Bode et al. (2020) reported increased mitochondrial fission in LA CMs derived from ZFS1 obese rats [194], and a multi-omics analysis revealed that the inflammatory response and mitochondrial fission were the main mechanisms responsible for myocyte stiffening in the DSSR model of HFpEF [195]. However, it remains to be elucidated whether the suggested fission abnormalities reflect the irregular muscle oxygen consumption, or if is a consequence of dysfunctional mitochondria triggered by mechanisms underlying HFpEF development, such as the increased mitochondrial oxidative stress.

### 5.3. Mitophagy

Mitophagy, the specific degradation of mitochondria by autophagy, is a fundamental process within the mechanisms of mitochondrial quality control. The turnover of dysfunctional mitochondria prevents the accumulation of impaired mitochondria, which is critical for cellular differentiation and for tissues with high energy demands, such as the cardiovascular system [148,196]. Several studies have already suggested that mitophagy inefficiency is linked to the progression of CVDs, like HF [197,198].

The mitophagy mechanisms and pathways have extensively been reviewed [199,200,201,202]. Briefly, the elimination of damaged mitochondria through mitophagy involves loading the damaged or dysfunctional mitochondria into autophagosomes via autophagy receptors like light chain-3 (LC3). The autophagosomes then fuse with lysosomes and the mitochondria are degraded by the activity of lysosomal enzymes [203]. Mitophagy can basically occur through ubiquitin-dependent or independent pathways. Ubiquitin-dependent pathways are normally triggered by the loss of mitochondrial membrane potential (MMP) and involve the PINK1/Parkin complex; ubiquitin-independent pathways, also called receptor-dependent pathways, involve BNIP3, FUNDC1, and lipid-dependent mechanisms [199,200].

Mitophagy is a central mechanism for mitochondrial quality and quantity control, through which dysfunctional mitochondria are degraded via autophagy to maintain homeostasis or, more specifically, mitohormesis [204,205,206,207]. It is well accepted that mitochondrial dysfunction impacts the progression and pathogenesis of HF; therefore, mitophagy is suggested to be cardioprotective [208]. In this sense, failing human hearts have reduced expression levels of autophagy-related proteins, such as Beclin1, LC3II, and PINK1 [209,210]. Mice with transverse aortic constriction (TAC)-induced HF show reduced PINK1 expression and Parkin-related mitophagy, as well as increased ROS production and CM apoptosis [211]. In addition, PINK1 ablation in mice compromises the mitochondrial translocation of Parkin and impairs mitochondrial homeostasis, thus facilitating CM apoptosis, myocardial fibrosis, and reduced capillary density [210]. These features reflect a HF phenotype, which is further evidenced by the onset of LVD and cardiac hypertrophy in young mice [210]. Nonetheless, these studies highlight that selective mild activation of PINK1-PRKN mitophagy in CMs could potentially prevent HF development and reduce cardiac injury.

Although recent studies suggest the role of autophagy in the development of HFpEF, the functional implications of mitophagy in HFpEF remain to be fully addressed. RNAseq analysis performed in HFpEF patients highlighted a potential link between endoplasmic reticulum stress, angiogenesis, and autophagy genes with the development of HFpEF [146]. In animal models, transcriptome analysis revealed that inflammation, extracellular matrix, endothelial function, sarcomere/cytoskeleton, apoptosis/autophagy, and those related to cell cycle and mitotic cell cycle processes were the most important upregulated pathways involved in HFpEF [212,213]. Moreover, it has been evidenced that mitophagy and autophagy are impaired in aged animal models [214,215]. Collaborating with these findings, spermidine, a natural polyamine, was found to enhance cardiac autophagy, mitophagy, and mitochondrial respiration; reduce inflammation; and prevent cardiac hypertrophy as well as diastolic dysfunction in HFpEF induced by DSSRs and HFD [216].

Taking all into account, it is increasingly clear that mitochondrial dynamics are crucial to health and the development of cardiac disease, with a belief that mitochondrial fusion is advantageous and fission is detrimental; however, this binary perspective may be overly simplistic. Fission is essential for the separation of damaged mitochondria for removal by mitophagy, and the exacerbation of myocardial injury may result from the excessive inhibition of this process. Therefore, it is imperative to achieve the appropriate equilibrium in the regulation of mitochondrial dynamics to facilitate potential clinical applications.

## 6. Mitochondrial Dysfunction and ROS Generation: Implications in HFpEF

### 6.1. Mitochondrial ROS Generation and Damage

Mitochondria complexes I and III are believed to be a major source of ROS. The superoxide radical (O_2_•−) is formed when a small percentage of electrons in the ETC prematurely reduce oxygen (O_2_) [217,218]. Under typical physiological conditions, ROS production comprises approximately 2% of the total oxygen consumed by mitochondria. Typically, an efficient antioxidant system situated within the mitochondrial matrix neutralizes the “excessive” mtROS [175,217] (Figure 5).

The first step in the neutralization process is conducted by a manganese-dependent superoxide dismutase (the mitochondrial isoform SOD2), which converts O_2_•^−^ to O_2_ and hydrogen peroxide (H_2_O_2_). The H_2_O_2_ is then reduced to water by glutathione peroxidase (GPX) and the mitochondrial isoforms of peroxiredoxin (PRX3 and PRX5) [219]. The regeneration of GPX and PRX is dependent on NADPH, whose reduction depends on TCA cycle activity. Therefore, to sustain the antioxidant systems and maintain ATP production, the TCA cycle integrity is needed. In this sense, mitochondrial dysfunction can disrupt the TCA cycle enzyme activity and OXPHOS, impairing ATP production and the redox balance, as characterized by variations in the NAD^+^/NADH, FAD/FADH2, and NADP^+^/NADPH ratios. The altered redox state then impacts the cell scavenger’s ability, leading to an increase in ROS release [147,220,221].

In mitochondria, there are numerous non-ETC sources of ROS. For instance, several mitochondrial flavoenzymes, such as α-ketoglutarate dehydrogenase, pyruvate dehydrogenase, glycerol phosphate, electron transfer flavoprotein (ETF), and ETF-oxidoreductase, are now recognized as significant ROS sources. In certain situations, they may produce a significantly greater quantity of O_2_•^−^ than ETC [221,222,223]. Moreover, during the FAO in the mitochondrial matrix, a significant amount of H_2_O_2_ is generated by flavoproteins situated upstream of complex III, such as short-, medium-, long-, and very long-chain acyl-CoA dehydrogenases (SCAD, MCAD, LCAD, and VLCAD, all belonging to essential enzymes for FAO) [224,225]. Furthermore, monoamine oxidase (MAO) in the OMM deactivates neurotransmitters such as serotonin and produces H_2_O_2_ as a secondary product [223,226]. Apart from mitochondria, there are other important cellular sources of ROS. The enzymes xanthine oxidase (XO), NADPH oxidase (Noxs), and uncoupled NOS also play a role in ROS production inside the cell [221,227].

It is widely known that ROS generation beyond the antioxidant system’s buffering capability contributes to oxidative stress, which can cause alterations in DNA, proteins, and lipids [223,228]. High concentrations of ROS cause non-specific damage to intracellular macromolecules, which typically generate more reactive intermediates and initiate a chain of damage multiplication [229]. Dysregulated ROS generation by FA results in both oxidative injury to the mitochondria and mitochondrial uncoupling, which in turn leads to impaired ATP production [230,231], suggesting a pathophysiological association between oxidative stress and mitochondrial dysfunction.

Mitochondria have their own circular, double-stranded DNA (mtDNA), which is 16,569 base pairs in length and encodes thirteen subunits of mammalian respiratory complexes, 22 tRNAs, and two rRNAs [232]. Compared to nuclear DNA, mtDNA is particularly vulnerable to oxidative stress. Numerous studies conducted on humans and animals have indicated that mtDNA may contain a substantially greater number of oxidized bases than nuclear DNA [232,233]. This is because mtDNA is in closer proximity to respiratory chains that produce ROS and lacks the protection of histone-like proteins observed with nuclear DNA. ROS (H_2_O_2_ and O_2_•^−^) generated in the mitochondrial inner membrane can easily react with proximal mtDNA, resulting in the accumulation of point mutations and/or deletions and a reduced DNA copy number. This process impairs mitochondrial functions [223,233,234].

Induced oxidative damage to mtDNA results in reduced amounts of mitochondrial transcripts and proteins, as well as impaired ETC performance and ATP generation. This, in turn, stimulates the creation of ROS in mitochondria and leads to mutations in mtDNA, so creating a self-perpetuating cycle of damage amplification [235,236]. Multiple investigations have shown that increased oxidative damage to mtDNA plays a role in the progression of cardiomyopathy and HF [237,238,239]. Heteroplasmy of mitochondria occurs when wild-type and ROS-mutated mtDNA coexist in the same mitochondrial structure. Once the proportion of aberrant mtDNA exceeds specific thresholds (60–70%), it disrupts mitochondrial homeostatic processes and reduces bioenergetic capability, mitophagy, and mitochondrial fusion and fission events. Cardiomyopathy develops when defective mitochondria are accumulated in CMs due to altered metabolic processes and communication pathways, reduced ATP generation, as well as cell shrinkage or death [240]. Since mitochondria can function as intracellular signaling centers that regulate nuclear gene expression at both the transcriptional and epigenetic levels, the rise in mitochondrial DNA heteroplasmy undeniably not only leads to malfunction in mitochondria but also triggers extensive changes in gene expression [241,242,243]. Thus far, more than 400 mitochondrial DNA abnormalities have been identified to be linked with human disorders such as HF [244].

Apart from mtDNA, lipids and proteins are also influenced by ROS. Lipid peroxidation may be elevated by the accumulation of lipid-derived metabolites and FA in the internal mitochondrial membrane. The development of metabolic cardiomyopathy is influenced by lipid peroxidation products, including malondialdehyde (MDA), 4-hydroxynonenal (4-HNE), and 4-hydroxyhexenal (4-HNE) [245]. There is evidence that lipid peroxidation products can damage mitochondrial membranes, suppress cyclooxygenase (COX), and activate uncoupling protein (UCP)-2, resulting in ETC dysfunction and an exacerbation of ROS production [245,246].

Cardiolipin, one of the primary phospholipids of the mitochondrial inner membranes, is also susceptible to modification by oxidants or ROS [247]. Oxidative modification of cardiolipin has significant effects on mitochondrial functions, such as opening mitochondrial permeability transition pores (mPTPs), reducing ETC activity, and impairing mitochondrial membrane fluidity [247]. Additionally, cardiac pathologies are associated with modifications to the mitochondrial proteome and associated mitochondrial dysfunctions [248]. It has been documented that the mitochondrial proteome is altered in response to cardiac oxidative stress [249,250]. In diabetic mice, TAC causes substantial cardiac mitochondrial proteome remodeling, which can be substantially reduced by the scavenging of mitochondrial ROS by mitochondrial catalase [250].

### 6.2. Mitochondrial ROS as Coordinators of Cellular Function and Mitohormesis

Historically, mtROS have been believed to be detrimental byproducts of cellular respiration, with a focus on aging and mitochondrial malfunction. However, recent studies have shown that mtROS are essential modulators of cellular signaling, adaptability, and stress responses rather than being hazardous byproducts [251]. Mitohormesis is the theory that low-to-moderate amounts of mtROS might activate protective pathways, which in turn improve cellular resilience and eventually sustain survival under hardship, potentially prolonging human life [204] (Figure 6). Mild stress caused by hypoxia, metabolic changes, or nutritional deprivation may induce the nucleus and cytoplasm to undergo metabolic and biochemical adaptations, which can assist cells in coping with more severe stressors in the future, rather than causing damage [204,217].

Studies in *C. elegans* have provided one of the most conspicuous examples of mitohormesis. Glucose deprivation raised mitochondrial respiration and ROS production, and the increased ROS triggered antioxidant defenses, promoting cellular adaptations, stress resistance, and increasing lifespan [252]. Pretreatment with antioxidants like N-acetylcysteine (NAC) prevented the hormetic response, highlighting ROS as active players in the adaptive processes that improve cellular function and longevity [252,253].

In line with this observation, mitohormesis has been associated with an extended lifespan in numerous organisms (Figure 6). For instance, the lifespan of Drosophila can be regulated by a redox-dependent mitohormetic response [254]. Similarly, in yeast, impaired TOR signaling increases mtROS production and extends the chronological lifespan [255]. Moreover, lifespan extension has been also observed in certain long-lived mutants, exemplified by *C. elegans* with impaired insulin/IGF-1 signaling, contingent upon mtROS [256]. In another instance, the activation of the hypoxia-inducible factor HIF-1 through the inhibition of mitochondrial respiration was triggered by oxidants [257]. A similar hormetic response, which involves the mtROS-dependent activation of HIF-1, was demonstrated to be crucial in the rescue of the AMPK-null mutant of *C. elegans* [258]. Therefore, the increased ROS, rather than being detrimental, was essential for the extension of lifespan, and mtROS seem to function as signaling molecules that induce adaptive responses, such as metabolic reprogramming and improved stress resistance.

Similarly, mtROS can function as second messengers that initiate cardioprotective signaling pathways under some specific circumstances. For example, the heart may be safeguarded from IR injury by mild oxidative stress that is induced by intermittent or moderate ROS production [259]. In CMs, modest concentrations of the mitochondrial redox cycler MitoParaquat (MitoPQ) were discovered to elevate mtROS levels without affecting mitochondrial function or calcium handling. In animal models, this moderate increase in ROS resulted in an increase in calcium accumulation in the sarcoplasmic reticulum, which in turn protected cells from anoxia/reoxygenation injury and reduced infarct size. However, excessive ROS induces loss of MMP, the opening of the mPTP, and altered calcium homeostasis [260]. This paradox exemplifies that at moderate levels, ROS can offer cellular defense; however, after reaching a threshold, increased ROS can impair mitochondrial and CM functions.

In addition to their function as protective messengers in CMs, mtROS have become essential regulators of cellular signaling [261]. ROS are essential messengers between mitochondria and other cellular organelles, particularly the nucleus. For instance, mtROS are involved in the regulation of macroautophagy [262], the modulation of hypoxia responses through HIF-1 [263], and the activation of cytosolic stress kinases [264]. MtROS can stimulate signaling pathways that promote cell survival and response to external stresses when the oxidative stress is moderate. Particularly, mtROS can activate the Nrf2 pathway, which enhances antioxidant defenses [265,266]. Experimental studies on *C. elegans* have shown that the initiation of defensive reactions, such as the AMPK pathway, relies on signaling that is dependent on mtROS [252,258].

Mitohormesis also underscores the reasons why antioxidant therapies, which are designed to decrease ROS levels, have been mainly unsuccessful in clinical trials [267,268,269]. Antioxidants may prevent the activation of protective hormetic pathways, impairing the stress management capacity. It is suggested that the increased mtROS production during physical exercise stimulates adaptative responses by the expression of antioxidant enzymes in the skeletal muscle [270]. Nevertheless, the advantageous effects of exercise are inhibited when individuals are administered antioxidant supplements (e.g., vitamin C and vitamin E) [270,271], suggesting that mtROS are required for the protective adaptative processes in response to exercise and other mild stressors.

Traditionally, mtROS have been considered hazardous agents in aging and disease. Additionally, the 1950 free radical theory of aging connected mtROS to cellular malfunction and to the development of several diseases such as cancer, diabetes, and dementia/neurodegeneration [272,273,274]. However, subsequent studies have contradicted this viewpoint, showing that mtROS are not only secondary consequences of mitochondrial malfunction but also essential regulators of cellular activity [222,251]. MtROS are now recognized as fundamental molecules for signaling biological pathways such as transcription, epigenetic, and stress responses [222]. Moreover, mtROS act as key mediators of cellular resilience and survival [275,276]. Recently, van Soest et al. have reported that mtROS released from mitochondria are unlikely to directly damage nuclear genomic DNA, even at extremely high levels [277]. These surprising results raise questions about the contribution of mtROS to chromosomal DNA mutations, and consequently to oncogenic transformation and aging.

Taking all into account, mtROS are more than just harmful respiratory byproducts. They act as regulatory signal molecules during metabolism, cell survival, and stress resistance. Mild-to-moderate ROS levels activate cellular defensive pathways, boosting cell resilience and longevity (Figure 6). Understanding the dual roles of mtROS in cellular signaling and damage is essential for developing targeted therapies that can mitigate disease progression while preserving necessary cellular functions.

### 6.3. Mitochondrial ROS and Heart Failure

As discussed in the previous section, mtROS are critical regulators of numerous intracellular responses through redox signaling pathways. Nevertheless, the function of mtROS in physiology or pathology is contingent upon their concentration, type, and the location of their formation [222]. The primary function of low levels of mtROS is to participate in physiological processes, including excitation–contraction coupling (ECC), cell proliferation, and differentiation. However, high levels of mtROS impair essential molecule structures and functions in CMs via oxidative stress-regulated signaling, which subsequently increases cardiac metabolism and malfunctions of mitochondria, ion channels, and transporters, as well as inflammation and CM death. Ultimately, these modifications lead to myocardial remodeling (hypertrophy, fibrosis, and apoptosis) and eventually HF [277,278] (Figure 6). 

Substantial mitochondrial remodeling, encompassing energy production, Ca^2+^ regulation, and the generation of ROS or reactive nitrogen species (RNS), has been well characterized during HF progression [279,280,281,282]. The subpopulations of mitochondria were first observed in the cell periphery, followed by interfibrillar mitochondria in HF [283]. Moreover, studies showed increased ROS generation during ischemic cardiac reperfusion [284,285], confirming that ROS can also enhance the generation of mtROS in pathological circumstances, leading to tissue death and inflammation.

CM apoptosis has a substantial impact on hypertrophic remodeling and cardiac dysfunction [286]. Multiple oxidative stress downstream signaling pathways have been implicated in CM apoptosis [287]. Previous investigations have demonstrated that the JNK signaling pathway is activated when human cardiac progenitor cells (CPCs) are exposed to H_2_O_2_, resulting in the inhibition of cell death [288]. The generation of ROS enhanced palmitate-induced apoptosis in CPCs, while the production of ROS was reduced by pretreatment with the NAC in human CPCs [289]. The activation of proapoptotic signaling pathways such as JNK, p38, and ASK1; the suppression of antiapoptotic signaling pathways like the PI3K/AKT pathway and the ERK1/2 pathway; and the immediate effects of ROS on mitochondria that trigger the release of cytochrome c may contribute to ROS-induced mitochondria-dependent apoptosis in CMs [248].

Substantial evidence of elevated ROS production in HF, which contributes to disease progression, has been obtained through extensive research in both human and animal models [290]. ROS imbalance was found in human ventricles explanted from patients with dilated cardiomyopathy. Using electro–paramagnetic resonance (EPR), the authors reported augmented mtROS production and increased myocardial oxidative stress. Further, enzyme activity assays revealed decreased activity of MnSOD in the mitochondrial matrix [291]. Moreover, the significance of mtROS in HF is reinforced by the phenotype of MnSOD knockout mice. Specifically, mice lacking MnSOD exhibit fatal dilated cardiomyopathy and HF, severe oxidative damage to the mitochondria and DNA, as well as a reduction in energy production. Additionally, mutant mice are exceedingly susceptible to hyperoxia, and the addition of exogenous ROS scavengers can suppress mitochondrial defects and attenuate cardiac remodeling [292,293]. Furthermore, new studies confirmed the causal involvement of mtROS imbalance in the pathogenesis of HF and offered mechanistic insight into the potential beneficial effects of antioxidants on HF [292,293]. Additional supportive evidence was recently provided using a guinea pig model of nonischemic HF [294]. In this study, the authors used a mitochondria-targeted ROS sensor to demonstrate an increase of mtROS production in quiescent and contracting LV myocytes from failing hearts, and they proposed that mtROS imbalance disrupts the optimal redox signaling that promotes the expression of genes involved in mitochondrial functions and ion handling [294].

Direct oxidation of histone deacetylase 4 (HDAC4) by elevated ROS can also activate hypertrophic signaling [295], which is linked to the nuclear export and de-suppression of transcription factors involved in cardiac hypertrophy, such as the myocyte enhancer factor 2 (MEF2) and calcineurin-NFAT [295,296]. The cardiac hypertrophic program can also be activated by chronic exposure to various agonists, such as AngII and endothelin-1, through the activation of NADPH-mediated ROS production [297].

ETC has recently garnered attention as a modulator of lipid peroxidation and GSH levels, which are responsible for ferroptosis (iron-dependent cell death) [298,299]. Contrary to complex III, complex I generated mtROS in isolated cardiac mitochondria, which promoted mitochondrial lipid peroxidation, as shown by the presence of lipid peroxy radicals [298]. The inhibition of both complex I and complex III greatly intensified the ferroptosis of CMs triggered by RSL3 (ferroptosis inducer) [300]. At the same time, the synthesis of GSH can safeguard cells from oxidative stress-induced cell death, including ferroptosis [301]. Despite the fact that mitochondria are unable to synthesize GSH, they roughly transport 10–15% of the total cellular GSH into the matrix via dicarboxylate carrier (DIC) and oxoglutarate carrier (OGC) [300]. Inhibiting DIC and OGC caused increased mtROS, membrane depolarization, and GSH depletion, as well as aggravated ferroptosis in CMs. Additionally, the ferroptotic PEox species were accumulated in mitochondria during cardiac I/R injury [300]. These data imply a causal role for mtROS in cardiac ferroptosis and dysfunctions.

It is worth noting that during the progression of HF, mitochondrial dysfunction can be both the cause and consequence of increased mtROS [302,303]. Enhanced mtROS is correlated with increased ATP generation in mitochondria, which is a result of increased ETC activity. Recent findings suggest that the acetylation of enzymes in failing hearts suppresses ATP synthase activity, in contrast to the notion that elevated energy demand in persistently stressed hearts can stimulate ATP production. When ATP generation is obstructed, it impairs ETC and increases mtROS emissions [304]. Further mtROS generation and more severe mitochondrial damage are the result of the initial increase in oxidative stress caused by mitochondrial injury [305,306]. MtROS are also implicated in a diverse array of cellular functions during HF. The opening of mPTP and the subsequent cell death are facilitated by increased mtROS. Both mtDNA damage and impaired mitochondrial biogenesis can be attributed to severe oxidative stress. The enzymatic activities inhibited by the oxidative alteration of proteins and lipids ultimately limit the capacity of mitochondrial respiration [307,308,309]. Moreover, inflammation is first triggered by the release of mtDNA. It has been recently demonstrated that mtROS function as essential mediators of inflammation, hypoxic signaling, the MAP kinase pathway, and retrograde communication between the nucleus and mitochondria in the context of HF [310].

Collectively, in HF, ROS imbalance can influence ETC coupling and mitochondrial bioenergetics, resulting in a vicious cycle that encompasses the oxidative modification of redox-sensitive kinases, cytosolic sodium accumulation, and further mtROS emission. This, in turn, can induce CM apoptosis, cardiac ferroptosis, arrhythmias, and cardiac remodeling through the post-translational oxidative modification of regulatory proteins, lipids, mtDNA, and nuclear DNA [311,312].

### 6.4. Mitochondrial ROS and HFpEF: A Fraction of the Whole?

Although studies on HFpEF and oxidative stress are scarcer, it has been suggested that oxidative and nitrosative stress could drive the onset and/or progression of HFpEF [5,84,313,314]. Chronic HF patients (CHF-NYHA functional class III) have increased levels of mtROS in circulating peripheral blood mononuclear cells (PBMCs) and worse mitochondrial functions when compared to mild HF patients (class I–II). MtROS generation in PBMCs is favorably linked with urine 8-hydroxydeoxyguanosine, a systemic oxidative stress measure, in CHF patients. Importantly, mtROS generation in PBMCs is highly associated with plasma levels of B-type natriuretic peptide, a diagnostic for HF severity, and inversely correlated with peak oxygen uptake, a measure of exercise capability, in HFpEF patients [315], suggesting a pathological role for mtROS in HFpEF (Figure 7).

NO and NOS are oxidized in the presence of ROS, resulting in a decrease in the bioavailability of NO [316], which controls mitochondrial pore opening [317]. Vasodilator NO is strictly regulated by hemoglobin and myoglobin, as well as O_2_ gradients in the myocardium to ensure that blood flow is appropriately adjusted to the metabolic requirements of the tissue. Specifically, NO production is reduced when O_2_ concentrations are high, and it is increased when O_2_ concentrations are low [317]. Under hypoxic conditions, NO reversibly binds to the O_2_ binding site on cytochrome c oxidase, thereby modifying the mitochondrial O_2_ consumption in tissues. The dysregulation of this mechanism in HF is linked to the progression of compensated to decompensated HF and impaired myocardial O_2_ consumption (MVO_2_) [318].

One of the most common comorbidities in HFpEF patients is diabetes, presenting in 40–45% of HFpEF patients [319,320]. The concurrent accumulation of glycative post-translational modifications has frequently been associated with the detrimental effects of oxidative stress on the development and progression of numerous CVDs, such as diabetic cardiovascular complications, atherosclerosis, and ultimately HF [321]. Glycation is the process by which a sugar non-enzymatically reacts with an amine group of a protein to produce advanced glycation end products (AGEs). These irreversible compounds compromise the structure and function of intra- and extracellular proteins, as well as binding their signal-transducing receptor (receptor for advanced glycation end products, RAGE) and causing a series of deleterious downstream effects [322]. It has been demonstrated that ROS increases the generation of AGEs, which in turn further enhances ROS production, thereby creating a detrimental positive feedback cycle of malignant AGE and ROS accumulation known as glycoxidative stress [321,323].

The most recent hypotheses on the cause of HFpEF are rooted in the increase of systemic inflammation and metabolic disruption, both of which are induced by non-cardiac comorbidities associated with HFpEF [324,325]. The change in the favored oxidative substrate, the rise in ROS levels, and defective mitochondria result in energy depletion due to an imbalance between ATP synthesis and demand. Several downstream signaling pathways are simultaneously activated, causing disruption to the normal heart architecture and initiating systemic inflammation that can lead to endothelial dysfunction of the coronary arteries and other unwanted events in HFpEF. Collectively, these processes stimulate the heart to undergo remodeling and hypertrophy in HFpEF, whereas, in HFrEF, the predominant cause is the loss of CMs mostly caused by ischemia, resulting in a weak, thin, or floppy heart muscle [67,326]. Previous studies have proposed that aberrant mitochondrial energy activity and oxidative stress occur before structural myocardial remodeling [327], implying a causal link between mtROS and cardiac dysfunctions.

Moreover, studies reported that the NLRP3 inflammasome can be assembled with mtDNA and N-formyl peptides, which can elicit inflammatory responses in mitochondria [328,329]. The activation of the NLRP3 inflammasome is now widely recognized as a critical link between mitochondrial dysfunctions and the integrity of the innate immune system [330,331]. During the progression of HF, cardiac inflammation is the result of increased mitochondrial injury caused by increased oxidative stress and mitophagy. It has been shown that mtDNA is released into the circulation during TAC surgery, and increased mtDNA is also detected in myocardial infarction patients, which activates circulating immune cells [332]. Additionally, the activation of NLRP3 can be influenced by mitochondrial redox conditions and the availability of NAD^+^ [333].

The critical function of mtROS as both a signaling molecule and a direct inducer of inflammation in HF has been underscored by recent research. Proinflammatory cytokines and chemokines induced by mtROS can worsen the inflammatory response [334]. Additionally, mtROS activates the NF-kB pathway, activating inflammatory genes and linking mitochondrial dysfunction to the inflammatory processes observed in HF [335]. During “Mito-inflammation”, the mitochondria-related inflammatory response, mitochondria function as both inducers of mitochondrial damage-associated molecular patterns (mtDAMPs) and the downstream effects of intracellular signaling pathways that are initiated by exogenous pathogen-associated molecular patterns (PAMPs) [336]. During mito-inflammation, mitochondria release mtDNA, ATP, mtROS, cardiolipin, and mitochondrial Ca^2+^ into cytosol or extracellular milieu in order to stimulate the expression and release of a variety of inflammatory mediators. MtROS may either directly promote oxidative damage to intraorganelle molecules and mtDNA or travel freely across the OMM. Upon its release into the cytosol, mtROS may activate proinflammatory signaling pathways, including the NF-kB, HIF, and AP-1 pathways, thereby contributing to the secretion of proinflammatory cytokines, such as IL-1 and IL-8 [331,337]. As recently proposed, the chronic activation of these proinflammatory pathways is linked to increased systemic inflammation and oxidative stress, causing capillary rarefaction and mitochondrial dysfunction, as well as the impairment of endothelium-dependent vasodilation. This, in turn, could contribute to abnormalities in the heart, vasculature, skeletal muscle, and other organs in HFpEF through a variety of pathways [21,338].

Taken together, emerging evidence suggests a bidirectional crosstalk between mitochondria metabolic stress and inflammation in the pathogenesis of the HFpEF disorder. Excessive mtROS production has a detrimental impact on mitochondrial function, resulting in a vicious cycle of increased ROS production, mitochondrial damage, and inflammation. MtROS can facilitate the release of mtDNA and other mtDAMPs, which can activate proinflammatory signaling pathways and further amplify inflammation as well as mtROS generation (Figure 7). The complexity of HFpEF is further exacerbated by the well-established heterogeneity among HFpEF phenotypes.

## 7. Therapeutics for HFpEF: Targeting mtROS and Associated Signal Molecules

Although the management of HFpEF continues to be a challenge, multiple treatments encompassing the management of comorbidities, consideration of non-pharmacological options, and implementation of guideline-directed medical therapy have currently been entering clinical trials, with mixed success (Table 2). A multidisciplinary approach is the optimal method, which would facilitate coordinated treatment plans to enhance quality of life and decrease hospitalizations by enabling early diagnosis and comprehensive management of HFpEF patients.

### 7.1. Conventional Pharmacological Therapy

#### 7.1.1. Diuretics

In the event of acute decompensation or to maintain euvolemia, diuretics should be administered to patients with HFpEF who have fluid retention to alleviate congestion and ameliorate symptoms [339]. Attention should be taken with obese patients that may have poor diuresis tolerance and a greater risk of renal dysfunction during decongestion [340].

#### 7.1.2. Sodium-Glucose Co-Transporter 2 (SGLT2) Inhibitors

SGLT2 inhibitors including dapagliflozin, empagliflozin, and sotagliflozin have been historically used in the treatment of type 2 diabetes, but emerging clinical evidence also shows that they also provide significant cardiovascular benefits in patients with HF. Specifically, in the DELIVER and EMPEROR-Preserved trials, the risk of HF hospitalization was reduced among individuals with HF and an ejection fraction ≥40% using dapagliflozin and empagliflozin [127,341]. In a meta-analysis that combined both trials, the SGLT2 inhibitors demonstrated a consistent reduction in the composite of CV-related death or first HF hospitalization, with HR 0.88, 95% CI 0.77–1.00 for CV-related death and HR 0.74, 95% CI 0.67–0.83 for HF hospitalization [342]. Sotagliflozin and empagliflozin were associated with improved outcomes, including mortality and HF events, among individuals admitted with acute decompensated HF in the SOLOIST-WHF and EMPULSE trials [343,344].

#### 7.1.3. Mineralocorticoid Receptor Antagonists (MARs)

There is evidence to suggest that MRAs may be beneficial in HFpEF [345]. In mice, the expression of proinflammatory factors in adipocytes was reduced with MRA treatment [346]. Nevertheless, the TOPCAT trial did not demonstrate a substantial advantage in a composite outcome of CV-related mortality, aborted cardiac arrest, or HF hospitalization among patients with HF and an ejection fraction of ≥45% [14]. However, a subsequent subgroup analysis of the TOPCAT trial revealed substantial regional variation in the treatment effect and outcomes, which was more favorable in patients with higher BMI [14,347]. Consequently, the MAR spironolactone is recommended for HFpEF at a grade of 2b according to the AHA/ACC/HFSA Guideline for the Management of HF [339].

#### 7.1.4. Angiotensin Receptor–Neprilysin Inhibitors

In comparison to valsartan (an angiotensin II receptor blocker), sacubitril–valsartan (a molecule combining angiotensin receptor blocking agent and neprilysin inhibitor) did not achieve a substantial decrease in CV-related mortality or HF hospitalization in individuals with HF and an EF ≥ 45% who were enrolled in the PARAGON-HF trial [348]. Nevertheless, a pre-specified subgroup analysis revealed that sacubitril–valsartan provided a benefit in individuals with LVEF below the median of 57% and in women in comparison to men. This benefit was primarily due to a decrease in HF hospitalization [349]. The FDA has recently approved sacubitril–valsartan for all patients with HF, with the benefits being most apparent in patients with a LVEF below normal. Recent expert consensus recommendations recommend that women with HF be treated with sacubitril–valsartan and spironolactone across the LVEF spectrum, unless otherwise contraindicated, due to the fact that women typically have higher EF than men because of their smaller size of the LV cavity [350,351].

#### 7.1.5. Angiotensin Receptor Blockers

The use of angiotensin receptor blockers alone can be interpreted as the establishment of a contraindication or other restriction on the use of angiotensin receptor–neprilysin inhibitors. In the CHARM-Preserved trial [352], treatment with an angiotensin receptor blocker candesartan was associated with a lower incidence of HF hospitalization compared to the placebo in patients with HF and an EF ≥ 40%. However, the I-PRESERVE trial did not demonstrate any observed benefit from the use of irbesartan, another angiotensin receptor blocker [13].

#### 7.1.6. β-Blockers

While β-blockers are not recommended for HFpEF, they are often used for other purposes, such as hypertension control, heart rate control, or in the treatment of angina in CAD. Recent meta-analysis has shown that β-blockers reduce all-cause and CV-related death in HFrEF and HFmrEF, but not in HFpEF [353]. Moreover, there is no consistent benefit from β-blockers in AF patients [353]. Potentially, β-blockers may worsen chronotropic intolerance in HFpEF, resulting in reduced exercise capacity. Specifically, β-blocker discontinuation in HFpEF patients with chronotropic incompetence can enhance maximal functional capacity, especially in those with reduced end-systolic LV volume [354]. However, a systematic review and meta-analysis of HFpEF studies showed that β-blocker therapy reduces all-cause mortality by 19% but has no impact on HF hospitalization or its composite with mortality [355].

### 7.2. Mitochondria Oxidative Stress-Targeted Therapy

Mitochondrion-targeted medicines have pros and cons over standard drugs. These medicines target mitochondrial dysfunction in HF, allowing for more precise cellular intervention and better outcomes [305,356]. By focusing on mitochondrial pathways, such drugs may avoid systemic side effects of traditional pharmaceuticals. Directly improving mitochondrial activity increases ATP synthesis, reduces oxidative stress, and improves cellular energy metabolism, improving heart function and patient health. However, developing and delivering mitochondrion-targeted medicines can be more complicated than standard pharmaceuticals, making it difficult to reach mitochondria safely and effectively. Although most of these medicines are targeted, off-target consequences are possible, especially if they affect mitochondrial function in noncardiac organs. In general, mitochondrial therapy acts by changing metabolic substrate utilization, improving OXPHOS, minimizing oxidative damage, or optimizing mitochondrial dynamics [357,358,359].

Regarding HF, evidence shows that the selective activation of the mitochondrial antioxidant system could protect against oxidative damage and the progression of cardiac dysfunction [294,360]. Indeed, mtROS-scavenging compounds are now undergoing human testing and have demonstrated good tissue permeability and subcellular targeting capabilities (e.g., NCT03506633, NCT02966665, NCT01925937, and ClinicalTrials.gov). Here we discuss some promising therapeutics targeting mitochondrial oxidative stress in the context of HF.

#### 7.2.1. Elamipretide

Elamipretide (Bendavia, MTP-131, SS31) is a mitochondria-targeted antioxidant. By preferentially binding to cardiolipin in the cytochrome c/cardiolipin complex, it optimizes mitochondrial electron transport, ATP generation and biogenesis [361,362]. Several animal models have demonstrated the efficacy of this water-soluble antioxidant, preventing doxorubicin-induced cardiotoxicity and mitochondrial damage in mice [363,364]. In failing hearts, the treatment with elamipretide was able to restore mitochondrial biogenesis and downregulate oxidation genes [365]. Moreover, post-MI rats treated with elamipretide showed improved cardiac functions, smaller infarct size, favorable pathological LV remodeling, and reduced ROS generation in non-infarcted areas [366]. In a canine model of microembolization-induced HF, elamipretide treatment at 0.5 mg/kg per day reduced LV dysfunctions, which was linked to a lower level of mtROS and better mitochondrial activity [367].

In addition to animal models, elamipretide has also been studied in patients. Elamipretide improves mitochondrial function in failing human hearts in vitro [368]. In HF patients, 4 weeks of elamipretide treatment was safe and well tolerated, but it did not improve LV end systolic volume [369]. Moreover, for MI patients, elamipretide was not able to restore heart function or improve cardiac functional measurements [370]. Therefore, the effects of elamipretide treatment in HF, particularly in HFpEF, need to be fully explored in more clinical trials.

#### 7.2.2. CoQ10

Coenzyme Q10 (CoQ10), a lipid-soluble quinone found in all cells, is crucial for electron transport and ATP production [371]. The benefits of CoQ10 in HF treatment are now widely accepted. In a 3-month study of 2664 NYHA class II and III patients, CoQ10 adjunctive therapy reduced adverse effects and improved clinical symptoms [372]. In addition, 420 HF patients receiving CoQ10 treatment (300 mg per day) for two years had improved NYHA class and HF survival [373]. Interestingly, CoQ10 seems to have longer positive effects on the heart. In a 5-year trial, selenium plus CoQ10 supplementation reduced CV mortality from 12.6% to 5.9% [374], and after a follow-up of twelve years, healthy participants and subgroups of patients with diabetes, hypertension, or ischemic heart disease on CoQ10 supplementation had a lower CV mortality risk [375]. This study shows that CoQ10’s protective effect on the heart lasts beyond the intervention period, proving its long-term efficacy in the heart.

#### 7.2.3. MitoQ

MitoQ (mitoquinone) is a CoQ (Coenzyme Q) derivative. MitoQ mimics natural CoQ and possesses high antioxidant action. MitoQ prevents superoxide-induced lipid peroxidation and mitochondrial damage [376]. Data from animal and human studies showed that MitoQ therapy prevents mitochondrial oxidative damage [377,378,379]. In a pressure overload animal model induced by ascending aortic constriction, MitoQ administration for 7 days reduced cardiac apoptosis, hypertrophic remodeling, and LV dysfunction via modulating mitochondrial-associated redox signaling [380]. Moreover, MitoQ was able to restore the cardiac mitochondrial network in HF animals [381] and reduce the hypoxia-induced cardiac dysfunctions in sheep and chicken [376]. Importantly, 6 weeks of MitoQ supplementation could improve vascular endothelial dysfunction and reduce aortic stiffness in older persons [382], suggesting a potential CV beneficial to aged people.

### 7.3. Non-Pharmacological Management

#### 7.3.1. Dietary Interventions, Low Carbohydrate Diet

In obese HFpEF patients, exercise tolerance and aerobic capacity are enhanced by nutritional or behavioral interventions that involve calorie restriction or weight loss [383,384]. Additionally, weight loss in obese HFpEF patients induced by gastric bypass surgery was associated with a regression of LV mass and wall thickness, as well as improvements in LV relaxation and LA filling pressure [383]. Moreover, a decrease in plasma sphingolipidome levels, implicated as a lower level of lipotoxicity, and an improvement in quality of life were also associated with surgery-induced weight loss [383]. Indeed, it was found that obesity, diabetes, and impaired cardiorespiratory capacity were associated with sugar consumption in patients with HFpEF [384]. The toxic effects such as cardiac dysfunctions, primarily affecting diastole, of high sugar consumption or Western diet were confirmed by experimental models and preclinical studies [385]. Normalizing the postprandial hyperglycemic index and insulinemic peaks by the modification/restriction of carbohydrate intake has been demonstrated to enhance all metabolic syndrome (MetS) manifestations and reduce CV risk [386]. Similarly, the consumption of unsaturated FAs (UFA) such as monounsaturated FAs (MUFAs) and polyunsaturated FAs (PUFAs) was correlated with improved cardiorespiratory capacity and diastolic functions, as well as increased fat-free mass in obese HFpEF patients, regardless of calorie intake. Similarly, mice fed a high-fat diet that was high in UFA and low in carbohydrates exhibited reduced weight gain and preserved myocardial functions [387]. In fact, UFA has been shown to suppress inflammation by reducing NLRP3 inflammasome activation in macrophages and in a type II diabetes model [388]. In addition, a low carbohydrate diet has been linked to lower rates of severe CV events and overall mortality in individuals with high CV risk, regardless of their body weight or overall dietary caloric intake [389]. Furthermore, a low-sugar diet can limit the generation of AGEs and, consequently, the deleterious glycoxidative stress [322]. Altogether, these findings highlight the capacity of a low-sugar/carbohydrate diet to modulate overall health and cardiac functions. They also demonstrate that the quality of nutrients, rather than the quantity of calories, could prevent CV events and control MetS manifestations.

#### 7.3.2. Exercise Training

In accordance with the concept of mitohormesis, exercise-induced oxidative stress leads to adaptive responses that enhance the endogenous antioxidant defense capacity and improve insulin sensitivity [270,271]. Exercise intolerance or dyspnoea upon exertion are important hallmarks of HFpEF. In this sense, exercise training is one of the few non-pharmacological interventions that has been successfully applied to HFpEF patients [390,391]. Exercise increases cardiorespiratory capacity measured by VO_2peak_ and quality of life in HFrEF and HFpEF [390,391]. The gain in VO_2peak_ and the improvement in physical capacity are linked to atrial reverse remodeling and improved LV diastolic functions in HFpEF patients [392]. The benefits of exercise in relation to VO_2peak_ appear not to be different between high-intensity intervals vs. moderate continuous training [393]. 

## 8. Conclusions

Despite the significant progress in our understanding over the past decades, HFpEF continues to be an intriguing condition for clinicians and researchers because it encompasses more than a single pathology, thereby presenting as a multifactorial and intricate disease. Therefore, a variety of mechanisms are expected to be involved in this pathology; however, an integrated understanding of HFpEF has yet to be achieved. ROS generated by mitochondria play dual roles in cellular physiology and pathology. Understanding the regulatory mechanisms of mtROS in HFpEF will be critical for developing precision therapies to address mitochondrial oxidative stress and its downstream effects. Nevertheless, due to the complicated nature of HFpEF, the current knowledge represents just a fraction of the whole. Further novel insights into the mtROS-inflammatory mechanisms contributing to HFpEF pathophysiology have the potential to benefit millions of individuals with HFpEF worldwide.

## Figures and Tables

**Figure 1 antioxidants-13-01330-f001:**
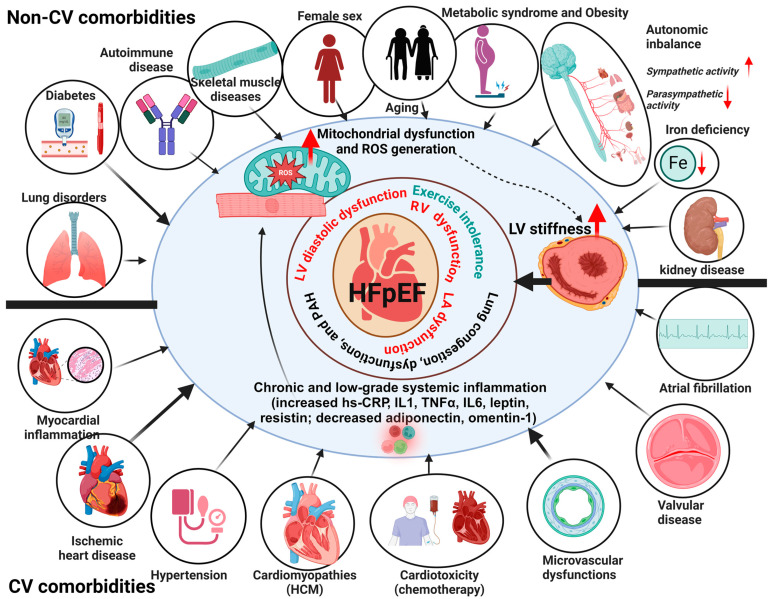
**Pathophysiology of HFpEF.** Multiple cardiovascular (CV) and non-CV comorbidities sequentially cause chronic and low-grade systemic inflammation, mitochondrial dysfunction, and ROS generation. These pathological alterations consequently trigger pathological remodeling and functional changes in the heart, which in turn increase LV stiffness, eventually causing HFpEF. HFpEF, heart failure with preserved ejection fraction; LV, left ventricular; NO, nitric oxide; ROS, reactive oxygen species. Red upwards and downwards arrows indicate increase and decrease, respectively. Diagram was created with URL (BioRender.com) (accessed on 27 October 2024).

**Figure 2 antioxidants-13-01330-f002:**
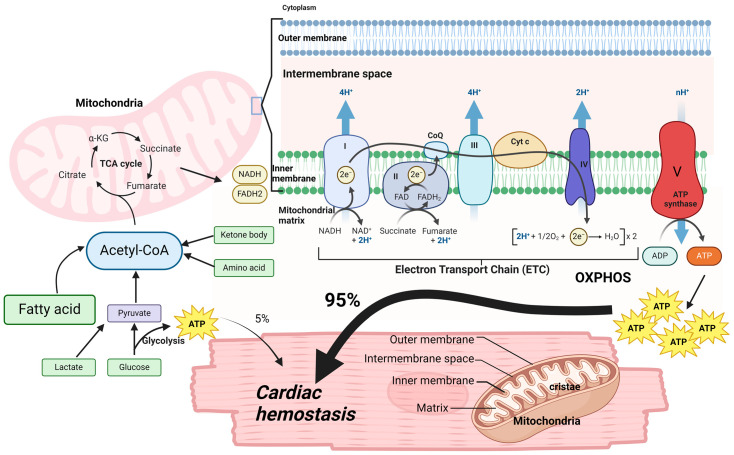
**Mitochondrial metabolism in cardiac hemostasis.** During cardiac hemostasis and under physiological conditions, the mitochondria in the heart produce 95% ATP by metabolizing a variety of fuels including fatty acids, glucose, lactate, ketones, pyruvate, and amino acids, primarily by mitochondrial oxidative phosphorylation (OXPHOS) through electron transport chain (ETC, or mitochondrial complex I–V). Tricarboxylic acid (TCA) cycle produces multiple metabolites (particularly NADH and FADH2), which enter ETC and function as electron carriers for ATP production. Additionally, 5% ATP is generated by anaerobic glycolysis. Diagram was created with URL (BioRender.com (accessed on 30 October 2024)).

**Figure 3 antioxidants-13-01330-f003:**
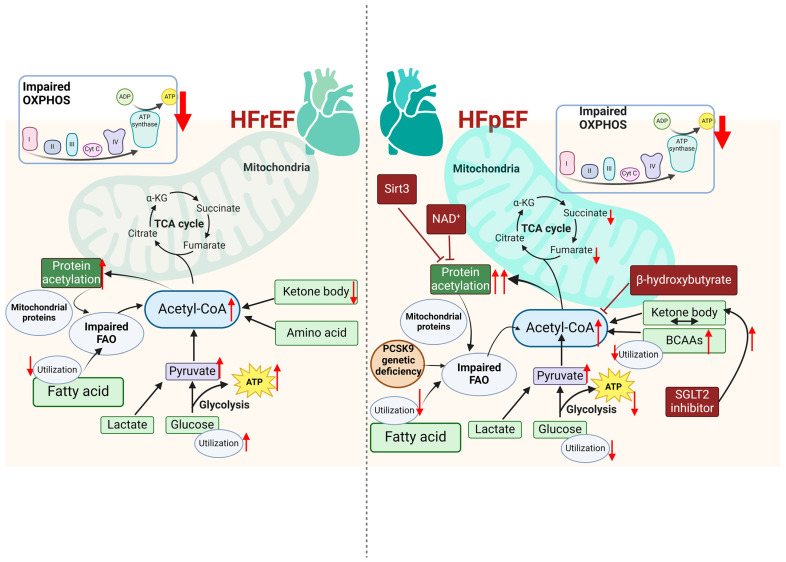
**Mitochondrial metabolism in HFrEF and HFpEF.** A healthy heart uses fatty acids (FAs) as its primary energy fuel to produce ATP through fatty acid oxidation (FAO), while glucose becomes the main energy fuel for ATP generation in heart failure via glucose oxidation. Although impaired mitochondrial oxidative phosphorylation (OXPHOS) is a common feature for both HFrEF and HFpEF, evidence suggests that the failing heart in HFpEF attempts to compensate for the impaired OXPHOS by increasing BCAA (branched chain amino acid) rather than glucose oxidation. The latter is predominately observed in HFrEF. Multiple molecules (such as Sirt3, NAD^+^, ketone body β-hydroxybutyrate, SGLT2 inhibitor) have been reported as the key regulators for mitochondrial metabolism, which could be the novel therapeutics for HFpEF. Interestingly, PCSK9 (proprotein convertase subtilisin/kexin type 9) genetic deficiency may cause HFpEF by impairing FAO, raising caution for the clinical applications of PCSK9 inhibitors. Red upwards and downwards arrows indicate increase and decrease, respectively. Diagram was created with URL (BioRender.com) (accessed on 27 October 2024).

**Figure 4 antioxidants-13-01330-f004:**
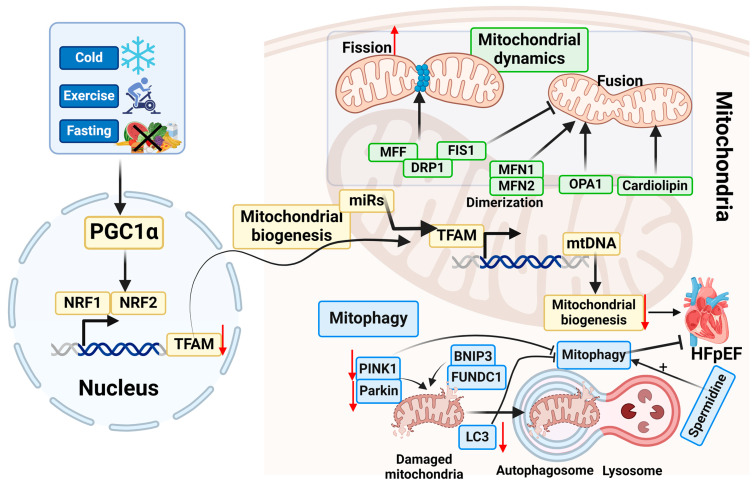
**Mitochondrial quality control (MQC).** MQC involved in mitochondrial biogenesis (new mitochondria generation), dynamics (fission and fusion), and mitophagy (degradation of the damaged or dysfunctional mitochondria) is a critical mechanism for protecting mitochondria and cellular functions. Decreased mitochondrial biogenesis, increased mitochondrial fission, and impaired mitophagy underscore HFpEF pathogenesis. Red upwards and downwards arrows indicate increase and decrease, respectively. Diagram was created with URL (BioRender.com) (accessed on 27 October 2024).

**Figure 5 antioxidants-13-01330-f005:**
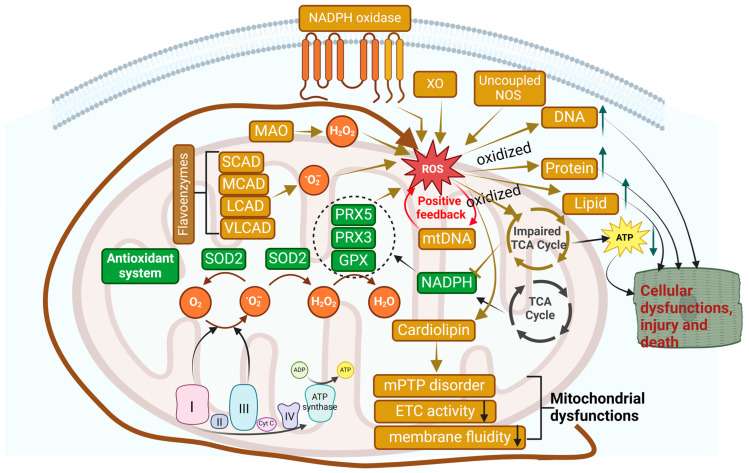
**Mitochondrial ROS generation and potential detrimental effects.** Although ROS can be generated by other cellular oxidative systems such as xanthine oxidase (XO), NADPH oxidase (Noxs), and uncoupled NOS, mitochondrial ROS (mtROS) generated from complexes I and III are the major source for cellular ROS. Cells including cardiomyocytes have an efficient antioxidant system situated within the mitochondrial matrix, neutralizing the “excessive” mtROS to fine-tuning cellular redox signaling. Imbalanced redox signaling generates excessive mtROS, which in turn oxidize DNAs, proteins, lipids, and mtDNAs and alter their corresponding functions, thereby causing a variety of cellular dysfunctions, injuries, and even death. Green upwards and downwards arrows indicate increase and decrease, respectively. Diagram was created with URL (BioRender.com) (accessed on 27 October 2024).

**Figure 6 antioxidants-13-01330-f006:**
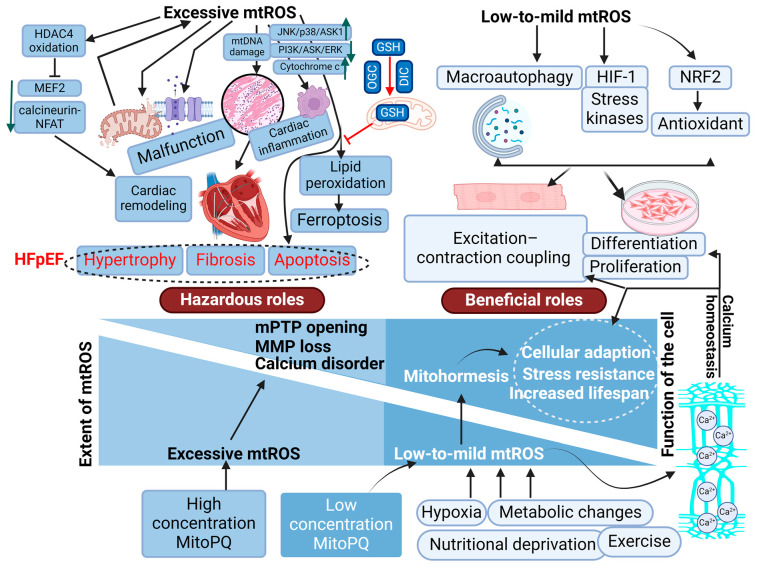
**Mitochondrial ROS in mitohormesis and mitochondrial dysfunction in cardiomyocytes (CMs).** Fine-tuning regulation of mitochondrial ROS (mtROS) is critical for cellular functions. While mitohormesis with low-to-mild levels of mtROS could activate protective pathways, improve cellular resilience against stress, and increase lifespan, excessive mtROS have detrimental effects on CMs such as hypertrophy, fibrosis, and apoptosis, eventually causing HFpEF. Green upwards and downwards arrows indicate increase and decrease, respectively. Diagram was created with URL (BioRender.com) (accessed on 27 October 2024).

**Figure 7 antioxidants-13-01330-f007:**
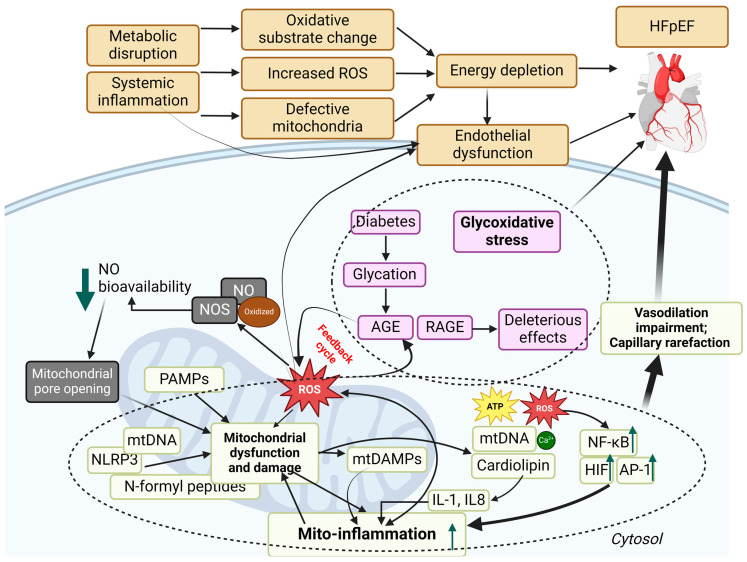
**Mitochondrial ROS in HFpEF.** In HFpEF, there is a bidirectional crosstalk between mitochondria metabolic stress and inflammation, namely “mito-inflammation”. Excessive levels of mtROS initiate a vicious cycle of increased mtROS production, mitochondrial damage, and inflammation. Increased mtROS promotes the release of mtDNA and other mtDAMPs, which in turn activate proinflammatory signaling pathways, amplifying inflammation and further increasing mtROS generation. In HFpEF with diabetes, high levels of glucose and increased mtROS trigger glycoxidative stress, a detrimental positive feedback cycle of malignant AGE and ROS accumulation. The resulting mito-inflammation causes endothelial dysfunction, vasodilation impairment, and capillary rarefaction, all contributing to HFpEF pathogenesis. Moreover, the changed oxidative substrates, increased ROS, and defective mitochondria due to metabolic disruption and system inflammation also cause energy depletion, underscoring their role in endothelial dysfunction and HFpEF. Green upwards and downwards arrows indicate increase and decrease, respectively. Diagram was created with URL (BioRender.com) (accessed on 27 October 2024).

**Table 1 antioxidants-13-01330-t001:** Animal models of HFpEF.

Animal Models										
	Hypertensive Phenotype						Cardio–Metabolic Phenotype			
Features	Rodent				Large Animal		Rodent			Large Animal
Model	DSSRs	SHRs	Aortic Constriction	Aldosterone Infusion	Aged Dogs + Perinephritis	Aortic-Banded Cats	ZSF1	Ageing	l-NAME + HFD	DOCA + Salt-Loaded Pigs + HFD
**Hypertension**	Y	Y	N	Y	Y	Y	Y	N	Y	Y
**Pulmonary Congestion**	N	N	Mild	Mild	-	Y	Y	-	Y	-
**Diastolic Dysfunction**	Y	Y	Mild	Y	Y	Y	Y	Y	Y	Y
**LVH**	Y	Y	Y	Y	Y	Y	Y	Mild	Y	Y
**Exercise Intolerance**	Y	Y	-	-	-	-	Y	Y	Y	Y
**Obesity**	N	N	N	N	N	N	Y	N	Y	Y
**Preserved LVEF**	Y *	Y *	Y *	Y *	Y	Y *	Y	Y	Y	Y

DOCA, deoxycorticosterone acetate; DSSRs, Dahl salt-sensitive; HFD, high-fat diet; l-NAME, N-nitro-l-arginine methyl ester; SHRs, spontaneously hypertensive rats; LVH, left ventricular hypertrophy; LVEF, left ventricular ejection fraction. * During the initial phase, the LVEF is preserved.

**Table 2 antioxidants-13-01330-t002:** Pharmacological and non-pharmacological management of HFpEF.

Therapeutics	Target(s)	Rationale	Major Findings/Outcome	Main Adverse Effects
**Non-pharmacological management**				
Dietary interventions; low carbohydrate diet	Adipose tissue (especially visceral fat), liver, and pancreas	Improve insulin sensitivity; reduce inflammation and glycoxidative stress	Improve MetS manifestations and cardiovascular risk	Risk of hypoglycemia in individuals on medication
Exercise training	Cardiac and skeletal muscles	Improve LV diastolic function and cardiac output	Improvement in VO_2peak_ peak and physical capacity	Muscle soreness, fatigue, and injury from over-exertion
**Conventional/repurposing drugs**				
Diuretics	Kidneys Cardiovascular system	Alleviate symptoms of fluid overload	Reduce congestion; control of blood pressure and pulmonary congestion	Electrolyte imbalance; hypotension
SGLT2 inhibitors	Kidneys	Reduce glucose and sodium reabsorption, leading to natriuresis and osmotic diuresis	Reduce fluid overload, inflammation, and oxidative stress; improve cardiac energy efficiency; reduce hospitalization and cardiovascular death	Volume depletion; electrolyte imbalance
Mineralocorticoid receptor antagonists	Mineralocorticoid receptors in the kidneys, heart, and blood vessels	Reduce sodium retention, lower blood pressure, and alleviate fluid overload; reduce myocardial fibrosis	Reduce hospitalizations; reduce myocardial remodeling	Hyperkalemia; gynecomastia; renal impairment
Angiotensin receptor–neprilysin inhibitors	Angiotensin II type 1 receptor and neprilysin	Reduce vascular stiffness and blood pressure; increase natriuresis	Reduce hospitalizations; reverse cardiac remodeling	Hypotension; hyperkalemia; angioedema
Angiotensin receptor blockers	Angiotensin II type 1 receptor/Renin–angiotensin–aldosterone system	Reduce the heart’s workload by lowering blood pressure	Reduce hospitalizations	Hypotension Hyperkalemia Renal dysfunction
β-Blockers	β-adrenergic receptors (heart and blood vessels)	Reduce heart rate and improve diastolic filling time	Slight reduction in hospitalizations; can improve exercise tolerance	Bradycardia; hypotension
**Mitochondria oxidative stress-targeted therapy**				
Elamipretide–Bendavia	Mitochondria	Improve mitochondrial bioenergetics; improve heart’s energy capacity	Improve mitochondrial function and cardiac remodeling	Injection site reactions
Coenzyme Q10	Mitochondria	Improve energy production; reduce oxidative stress	Improve LVEF and exercise intolerance; may reduce hospitalization	Gastrointestinal issues
MitoQ: mitoquinone	Mitochondria	Neutralize oxidative stress; restore mitochondrial function	Enhance mitochondrial respiration; may improve exercise tolerance and reduce fatigue	Gastrointestinal issues

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
