# Peer review of "Mitochondrial Reactive Oxygen Species Dysregulation in Heart Failure with Preserved Ejection Fraction: A Fraction of the Whole"

_antioxidants, 2024, doi:10.3390/antiox13111330_

Round 1
Reviewer 1 Report
The present manuscript is an interesting study of current relevance, but several issues must be addressed to improve its quality
Line 94: Reconsider citations 13 and 14 as they seem out of context.
Line 101: Correct the typo; it should be "Olmsted County" instead of "country."
Lines 127-129: Rephrasing is needed as the current phrasing distorts the meaning of the sentence.
Line 190: Correct the repetition of "right RVD" (since "R" already stands for "right").
Lines 202-204: Rephrase the sentence because it is confusing in its current form. Carbon monoxide exchange capacity is just one method to measure lung diffusing capacity, which is actually decreased in HFpEF.
Lines 540-541: Rephrase to clarify that PCSK9 inhibitors are used to treat hypercholesterolemia and to reduce cardiovascular risk in patients with ischemic heart disease, but they are not directly used to treat ischemic heart disease.
References: Several citations have spelling errors or are not properly formatted (e.g., citation 10).
Author Response
Remarks to the Author:
The present manuscript is an interesting study of current relevance, but several issues must be addressed to improve its quality.
Response: Thank you very much for your positive and encouraging comments about our work. We hope that we have addressed all your concerns/criticisms in the revised manuscript.
Detail comments:
- Line 94: Reconsider citations 13 and 14 as they seem out of context.
Response: Removed these two citations from the revised manuscript as suggested.
- Line 101: Correct the typo; it should be "Olmsted County" instead of "country."
Response: Thanks for pointing out this typo, which has now been corrected in the revised manuscript (Line 98).
- Lines 127-129: Rephrasing is needed as the current phrasing distorts the meaning of the sentence.
Response: we have removed this sentence from the revised manuscript as suggested (Line 124).
- Line 190: Correct the repetition of "right RVD" (since "R" already stands for "right").
Response: Thanks for pointing out this repletion, which has now been corrected in the revised manuscript (line 183).
- Lines 202-204: Rephrase the sentence because it is confusing in its current form. Carbon monoxide exchange capacity is just one method to measure lung diffusing capacity, which is actually decreased in HFpEF.
Response: Thank you very much for your suggestion. We have now revised this sentence in the revised manuscript (line 194-196).
- Lines 540-541: Rephrase to clarify that PCSK9 inhibitors are used to treat hypercholesterolemia and to reduce cardiovascular risk in patients with ischemic heart disease, but they are not directly used to treat ischemic heart disease.
Response: Thank you very much for your suggestion. We have now revised this sentence in the revised manuscript (line 496-497).
- References: Several citations have spelling errors or are not properly formatted (e.g., citation 10).
Response: As suggested, we have carefully checked all the citations to ensure their accuracy with proper format.
Reviewer 2 Report
The manuscript is a narrative review that in the intentions of the authors focuses on the role of mitochondrial dysfunction in the pathogenesis of HFpEF.
Overall the paper is interesting. However there are some points that should be iimproved
Firstly the style used seems too scholastic to me and seems to be written for students
I find the manuscript readable but dispersive: too much space is dedicated to the epidemiology and pathophysiology of HFpEF. I suggest merging and significantly reducing the size of the first 3 paragraphs.
In the central part of the manuscript, dedicated to mithocondrial function, references to biochemistry appear to be redundant and excessive. Please semplify the text ,
I suggest to avoid the paragraphs dedicated to pharmacological therapy of HFpEF as it is not relevant to the topic of the review.
Figure 1: I find this figure too confusing . Please, simplify it.
Author Response
Remarks to the Author:
The manuscript is a narrative review that in the intentions of the authors focuses on the role of mitochondrial dysfunction in the pathogenesis of HFpEF. Overall, the paper is interesting. However, there are some points that should be improved.
Response: We are grateful for all constructive recommendations and positive comments that have helped us to improve our manuscript. Please see our point-by-point response detailed below and included in the revised manuscript.
Detail comments:
- Firstly, the style used seems too scholastic to me and seems to be written for students. I find the manuscript readable but dispersive: too much space is dedicated to the epidemiology and pathophysiology of HFpEF. I suggest merging and significantly reducing the size of the first 3 paragraphs.
Response: As advised, we have now rewritten this section.
- In the central part of the manuscript, dedicated to mithocondrial function, references to biochemistry appear to be redundant and excessive. Please simplify the text.
Response: Thanks for your recommendation. We have now simplified this section as suggested (section 4.1).
- I suggest to avoid the paragraphs dedicated to pharmacological therapy of HFpEF as it is not relevant to the topic of the review.
Response: We are grateful for your thoughtful suggestion. However, summarising and discussing potential pharmacological therapies for HFpEF is one of our primary objectives in this review, which would also benefit to our readers in our opinion. Accordingly, we decided to keep this chapter in our revised manuscript.
- Figure 1: I find this figure too confusing. Please, simplify it.
Response: Thank you very much for your suggestion. We have revised Figure 1 to make it simper.
Reviewer 3 Report
The authors (Martinez C.S. et al. (Mitochondrial reactive oxygen species dysregulation in heart failure with saved ejection fraction: a fraction of the whole) present a review of the evidence for the involvement of mitochondrial dysfunction in chronic heart failure particularly with an ejection fraction (EF) >50%. The article presents a thorough but educational review with 8 chapters and 404 references. The comparison of preserved EF to reduced EF is obvious and is clearly presented in 3 figures. This review is of interest for both researchers and clinicians.
Minor points.
The references need to be reviewed in terms of presentation.
Table 1 only presents animal models for preserved EF, all associated with diastolic dysfunction only during an initial phase. This phase must be clearly defined in terms of period.
Do you have information on aged SHR for EF? To my knowledge other species have been employers (rabbits, ferrets, guinea pigs…).
The use of rodents is different in terms of heart rate (> 280 BPM) and heart rate variability from human and large animal models, do you have any information. Other animals reduction models could also present a preserved initial phase of EF.
No information or limited facts are presented in terms of therapeutic or preventive pharmacological intervention in these models?
Animal models are limited because most of them have presented beneficial effects of many drugs but not demonstrated in humans, this is the case for HFrEF and inotropes, antiarrhythmic drugs. What is the state of the art in terms of therapeutics for animal models of HFrEF?
Conventional drugs for HFrEF presented in table 2 have been developed for different purposes (cardiovascular, renal, etc.), Bendavia as a new peptide seems important to reduce hospitalizations and quality of life, do we have results in terms of mortality?
A table with the inappropriate drug (used for HFrEF) and the reason is appreciated.
Impaired but silent diastolic dysfunction could be a cause of development of comorbidities?
Author Response
Remarks to the Author:
The authors (Martinez C.S. et al. (Mitochondrial reactive oxygen species dysregulation in heart failure with saved ejection fraction: a fraction of the whole) present a review of the evidence for the involvement of mitochondrial dysfunction in chronic heart failure particularly with an ejection fraction (EF) >50%. The article presents a thorough but educational review with 8 chapters and 404 references. The comparison of preserved EF to reduced EF is obvious and is clearly presented in 3 figures. This review is of interest for both researchers and clinicians.
Response: Thank you for your positive comments about our manuscript, which help us to improve this review.
Minor points:
- The references need to be reviewed in terms of presentation.
Response: As suggested, we have now carefully checked all the references to avoid any mis-presentation or mis-formatting.
Detail comments:
- Table 1 only presents animal models for preserved EF, all associated with diastolic dysfunction only during an initial phase. This phase must be clearly defined in terms of period. Do you have information on aged SHR for EF? To my knowledge other species have been employers (rabbits, ferrets, guinea pigs…).
The use of rodents is different in terms of heart rate (> 280 BPM) and heart rate variability from human and large animal models, do you have any information. Other animals reduction models could also present a preserved initial phase of EF.
No information or limited facts are presented in terms of therapeutic or preventive pharmacological intervention in these models?
Animal models are limited because most of them have presented beneficial effects of many drugs but not demonstrated in humans, this is the case for HFrEF and inotropes, antiarrhythmic drugs. What is the state of the art in terms of therapeutics for animal models of HFpEF?
Response: Thank you very much for your constructive suggestion. The main animal models used to simulate human HFpEF were summarised and included into Table 1 and Chapter 3.
- Actually, we clearly indicated that some animal models of HFpEF only simulate HFpEF phenotype during the initial phase in Table 1 (as indicated with ‘*’).
- We fully agreed with the review that although enormous efforts have been put into establishing an ideal animal model to best simulate all the features and characteristics of human HFpEF, there are still numerous differences between human HFpEF and all the animal models currently used to mimic HFpEF, which has been mentioned and discussed in our review (Line 238-245, 290-302).
- Indeed, many animal models based on the “Hypertensive phenotype” used to study HFpEF have a common limitation, that is the preserved LVEF occurred only during the initial phase of the disease. Specifically, in the case of aged SHRs, they frequently exhibit concentric hypertrophy and HFpEF over 16–32 weeks, with intact systolic function and a prolonged LV isovolumic diastolic period. However, more than half of the animals will progress to systolic dysfunction after 12 months (Holjak et al. Am J Hypertens. 2022; Marzak et al. J Hypertens. 2014; Wang et al. Evid Based Complement Altern Med. 2017).
- On the other hand, some animal models (ZSF1, Ageing, l-NAME + HFD, DOCA + salt-loaded pigs +HFD) based on a “Cardio-metabolic phenotype” present normal values for LVEF and develop diastolic dysfunction progressively, without apparent systolic dysfunctions for long term (over 15 weeks), presenting a high similarity to the human disease.
- Regarding the pharmacological intervention in animal models of HFpEF, for the models based on the ‘Hypertensive phenotype’, the use of angiotensin II type 1 receptor blocker (ARB) has been reported to reduce left ventricular hypertrophy, fibrosis and diastolic dysfunction even at an advanced stage of hypertensive diastolic HF (Nishio et al, J Hypertens 2007, Yamamoto et al, Cardiovasc Res 2000). However, such therapeutical effects of ARB are absent in the human disease. In metabolic HFpEF preclinical models, SGLT2 inhibitors are the state of the art in terms of therapeutics. In cardio-metabolic models, SGLT2 inhibitors have been shown to enhance ATP production rates and glucose metabolism, hence ameliorating cardiac dysfunction and avoiding the onset of HFpEF (Zhang et al Cardiovasc Diabetol. 2019; Withaar et al. Cardiovasc Res. 2021). Accordingly, SGLT2 inhibitors have been recently approved for treating HFpEF clinically. All the information regarding animal models and therapeutics were discussed in Chapter 3 and 7 in this review.
- Conventional drugs for HFrEF presented in table 2 have been developed for different purposes (cardiovascular, renal, etc.), Bendavia as a new peptide seems important to reduce hospitalizations and quality of life, do we have results in terms of mortality? A table with the inappropriate drug (used for HFrEF) and the reason is appreciated.
Response: Thanks for your thoughtful suggestion. However, discussing the inappropriate drugs used for HFrEF and associated reasons is beyond the main purpose of this review article.
Indeed, Bendavia (Elamipretide) is a mitochondria-targeting peptide that is currently being studied for its potential benefits for HF patients, with a particular emphasis on the improvement of mitochondrial function and the reduction of oxidative stress. However, to our knowledge, information related to the mortality rate in HFpEF patients treated with Bendavia is still lacking.
- Impaired but silent diastolic dysfunction could be a cause of development of comorbidities?
Response: Thanks for pointing out this issue. We fully agreed with the review that impaired but silent diastolic dysfunction could be a cause of development of comorbidities. However, such scientific evidence is still lacking. Indeed, there are still a long way to go until we have the full picture of the pathophysiology of HFpEF, and a deeper understanding into the cross-talking between heart and other organs/tissues in the context of HFpEF.